# Physics-aware Machine Learning for Glacier Ice Thickness Estimation: A Case Study for Svalbard

Viola Steidl[1], Jonathan L. Bamber[1,2], and Xiao Xiang Zhu[1,3]

[1]Chair of Data Science in Earth Observation, Department of Aerospace and Geodesy, Technical University of Munich, 80333 Munich, Germany
[2]Bristol Glaciology Centre, School of Geographical Sciences, University of Bristol, Bristol, BS8 1SS, UK
[3]Munich Center for Machine Learning, 80538 Munich, Germany

**Correspondence:** Viola Steidl (viola.steidl@tum.de)

**Abstract.** The ice thickness of the world's glaciers is mostly unmeasured, and physics-based models to reconstruct ice thickness can not always deliver accurate estimates. In this study, we use deep learning paired with physical knowledge to generate ice thickness estimates for all glaciers of Spitsbergen, Barentsøya, and Edgeøya in Svalbard. We incorporate mass conservation and other physically derived conditions into a neural network to predict plausible ice thicknesses even for glaciers without any in situ ice thickness measurements. With a glacier-wise cross-validation scheme, we evaluate the performance of the physics-informed neural network. The results of these proof-of-concept experiments let us identify several challenges and opportunities that affect the model's performance in a real-world setting.

## 1 Introduction

Glacier ice thickness is a fundamental variable required for modelling the evolution of a glacier. Ice thickness is the most important input for modelling the dynamics of an ice mass because surface velocity is proportional to the fourth power of thickness (Cuffey and Paterson, 2010). Combined with surface elevation, it provides bed topography, also key for modelling flow (van der Veen, 2013). However, direct measurements of ice thickness are scarce. In situ ice thickness measurements exist for only a fraction of the 215 000 glaciers in the world (Welty et al., 2020).

There are physics-based and process-based approaches that aim to reconstruct glacier ice thicknesses from in situ data and ice dynamical considerations. Farinotti et al. (2017) compared 17 models and found that their ice thickness estimates differ considerably on the test glaciers. Following these results, Farinotti et al. (2019) created an ensemble of five models to develop a *consensus* estimate of ice thicknesses for the world's glaciers in 2019. Later, Millan et al. (2022) derived ice thickness estimates for the world's glaciers using ice motion as the primary constraint. However, these results still differ from Farinotti et al. (2019) *consensus* estimate. It is evident, therefore, that significant uncertainty remains in ice thickness estimates.

Machine learning approaches are flexible and adapt well to complex structures and non-linear behaviour. They have already been employed to model glacier quantities like surface-mass-balance (Bolibar et al., 2020; Anilkumar et al., 2023) or ice thickness (Haq et al., 2021; Leong and Horgan, 2020), classify surge-type glaciers (Bouchayer et al., 2022), or model glacier flow (Jouvet, 2023; Min et al., 2019). One advantage of machine learning approaches is their efficient optimization and evaluation

compared to process-based models (Jouvet et al., 2022). The disadvantages of purely data-driven machine learning models are that they do not guarantee the physical correctness of the predicted quantities, and they often need huge amounts of training data to fully represent the system's complexity.

Recently, a new framework of data-driven but physically constrained models was described as physics-informed neural networks (PINNs) by Raissi et al. (2018). They exploit that neural networks can represent solutions to partial differential equations (PDE) if the squared residual of the governing PDE acts as the loss function of the neural network (Lagaris et al., 1998). Partial derivatives with respect to the model inputs are easy to calculate with the automatic differentiation algorithm that is used to train neural networks. PINNs do not require a discrete grid to be evaluated. Therefore, the physics-based loss can be evaluated at any point within the training domain (Xu et al., 2023). Additional ground truth data can be used to compute a data loss that acts as an internal condition to constraining solutions to the PDE.

PINNs and variations thereof were also already used for predicting ice flow (Jouvet and Cordonnier, 2023), inferring basal drag of ice streams (Riel et al., 2021) or ice shelf rheology (Wang et al., 2022; Iwasaki and Lai, 2023), for example.

Cheng et al. (2024) built a unified framework involving a PINN to model ice sheet flow by enforcing momentum conservation derived from the Shelfy-Stream Approximation. They apply their framework to a single glacier in Greenland to showcase the ability of the PINN to reconstruct ice thickness and basal friction simultaneously.

Instead of using momentum conservation, Teisberg et al. (2021) created a mass-conserving PINN to produce realistic ice thickness and depth-averaged ice flow maps for a single glacier in Antarctica. They showed that, for their case, solving for mass conservation and additional constraints produces a valid ice thickness estimate.

This work extends the mass-conserving approach to predict ice thickness for multiple glaciers. As a proof of concept, we include all non-surging glaciers in Spitsbergen, Barentsøya, and Edgeøya in Svalbard to show that it is possible to use a PINN architecture for a heterogeneous region. These regions include glaciers with various morphologies, from valley glaciers to ice caps. To better account for the variety of glacier geometries, sizes, and flow velocities, we include additional input features, e.g., slope and elevation. The challenge is to find a configuration of inputs and physical constraints that is general enough to describe the variety of glaciers in the study region. At the same time, the constraints and inputs should be strict enough to force the model to produce physically correct outputs.

Ice thickness measurements exist only for a fraction of the glaciers in the dataset, and there is no benchmark dataset to measure the model's performance. Therefore, we need a validation method that assesses the performance, although there is no ground truth. To this end, we estimate the expected drop in performance for glaciers without ice thickness measurements performing glacier-wise cross-validation. Also, we compare our ice thickness estimates to those of Millan et al. (2022), the *consensus* estimate of Farinotti et al. (2019) for our study region. These estimates are no benchmark datasets but are widely accepted in the community. Additionally, we compare to the recently published ice thickness estimate of van Pelt and Frank (2024) tailored to the region of Svalbard.

Finally, we discuss the challenges and opportunities of the approach and pathways to improve the ice thickness estimate.

## 2 Physics-aware machine learning

### 2.1 Mass conservation

Assuming ice to be incompressible and integrating vertically along the depth of a glacier, we retrieve the two-dimensional form of the mass conservation

$$\frac{\partial H}{\partial t} + \nabla \cdot (\bar{\boldsymbol{v}} H) = \dot{b} \tag{1}$$

with $H$ being the ice thickness and $\dot{b}$ denotes the mass balance of the glacier. $\bar{\boldsymbol{v}} = (\bar{v}_x, \bar{v}_y)$ is the velocity caused by the deformation of ice, averaged along the vertical axis. We will refer to $\bar{\boldsymbol{v}}$ as the depth-averaged velocity in the following. Equation (1) can be reformulated as

$$\nabla \cdot (\bar{\boldsymbol{v}} H) - \dot{a} = 0 \tag{2}$$

with $\dot{a} = \dot{b} - \partial_t H$ known as the apparent mass balance. In other words, the flux divergence on a glacier equals its apparent mass balance.

### 2.2 Depth-averaged velocity and basal sliding

Glacier flow is the result of gravity-induced stresses on the ice. Friction between the ice and the glacier bed or sidewalls, friction between slower and faster-moving ice within the glacier, and gradients in longitudinal tension or compression balance the gravitational stress (van der Veen, 2013). The resulting ice movements depend on many factors, such as the physical properties of the ice like temperature, impurities, or density, and also conditions at the glacier bed (Jiskoot, 2011).

From space, we can observe the surface velocity of glaciers. To infer thickness from mass conservation we would need to know the depth-averaged velocity.

There are models with different degrees of approximations to the full Navier-Stokes equations to describe ice flow. The simplest one, the shallow ice approximation (SIA) assumes lamellar flow, so the driving forces are entirely opposed by basal drag. It neglects lateral shear and longitudinal stresses. The rate factor A from Glen's flow law is taken to be constant with depth (van der Veen, 2013).

From this model, we can derive that the depth-averaged velocity relates to the surface velocity like $\bar{v} = 0.8 v_{\mathrm{s}}$ assuming the flow velocity at the base of the glacier is 0 (see Appendix A for derivation). However, basal velocity is unlikely to be 0.

The basal sliding velocity tightly relates to the properties of the glacier bed and complex interactions between water, sediment, and ice at the glacier bed (Cuffey and Paterson, 2010). Millan et al. (2022) introduced an empirical factor $\beta$ with $v_{\mathrm{b}} = \beta v_{\mathrm{s}}$ to account for contributions from basal sliding. They derive the factor from the ratio between surface slope and surface velocity.

If the ice velocity is entirely by slip along the glacier bed then $v_{\mathrm{s}} = v_{\mathrm{b}} = \bar{v}$. Accordingly, we estimate the depth-averaged velocity to be within the bounds of

$$(l_{\mathrm{lower}} + (1 - l_{\mathrm{lower}}) \cdot \beta) \cdot v_{\mathrm{s}} < \bar{v} \leq v_{\mathrm{s}} \tag{3}$$

where $l_{\text{lower}}$ acts as a parameterization for the vertical integration of the velocity and can be set between 0 and 1. Depending on the factor $\beta$ that lies between 0.1 and 1 the lower boundary is close to the defined $l_{\text{lower}}$ or closer to 1. For $\beta = 1$ the lower boundary for the depth-averaged velocity equals the surface velocity.

Depth-averaged velocities are estimated for the x- and y-direction and the velocity magnitude. Therefore, we calculate three separate $\beta$-values.

## 2.3   PINN Model

As already mentioned, a PINN consists of a neural network that is able to approximate the solution to a PDE (Karniadakis et al., 2021). A neural network, also sometimes called multi-layered perceptron, consists of layers of connected nodes, also

called neurons, where the connections each have an associated weight. At each node, the weighted outputs from each node of the previous layer are passed through a non-linear activation function (Goodfellow et al., 2016). By minimizing a loss, the weights of the network are updated to make accurate predictions.

     In a PINN model the loss is given by the residual of the PDE we want to solve. In theory, PINNs only require input features that are needed to calculate the terms in the PDE (Raissi et al., 2018). In our work, we also provide the neural network with

auxiliary data, that is related to glacier ice thickness but is not needed to solve the PDE. Therefore, we can exploit information from observable data as we would do it with a non-physics-aware neural network.

     Additionally, we use a Fourier feature encoding layer as described by Tancik et al. (2020) preceding the neural network. A Fourier feature encoding layer maps input vector **x** to a higher dimensional feature space using

$$\gamma(\mathbf{x}) = [\cos(2\pi\mathbf{B}\mathbf{x}), \sin(2\pi\mathbf{B}\mathbf{x})]^{\mathsf{T}}. \tag{4}$$

The embedding of spatial coordinates was originally developed to overcome spectral bias in neural networks and speed up convergence in the reconstruction of images. It enables the network to learn high-frequency functions in low-dimensional problem domains.

     Figure 1 shows a schematic of the PINN model with its input features, outputs, and loss components. The exact architecture of the PINN is described in Appendix B. The inputs to the model are vectors for each grid cell in the study region. They contain

the spatial coordinates and surface velocities in x- and y-directions, and three $\beta$ values to correct for basal sliding in x- and y-direction and in the magnitude. Additionally, the vectors contain auxiliary data like elevation, slope, the grid cell's distance to the border of its glacier, and the area of the glacier it belongs to.

     Only the spatial coordinates get mapped to higher dimensional Fourier features.

     The model outputs three quantities at each point of query: the ice thickness $H$ and depth-averaged velocity $\bar{v}$ in x- and

y-direction. The predicted quantities must fulfil the mass conservation described in Eq. (2). The squared deviation from this equation is the first component of the loss function:

$$\mathcal{L}_{mc} = (\nabla \cdot (\bar{\boldsymbol{v}}H) - \dot{a})^2 \tag{5}$$

     The second component of the loss function is the amount by which the depth-averaged velocity estimates in the horizontal plane exceed the boundaries given in Eq. (3):

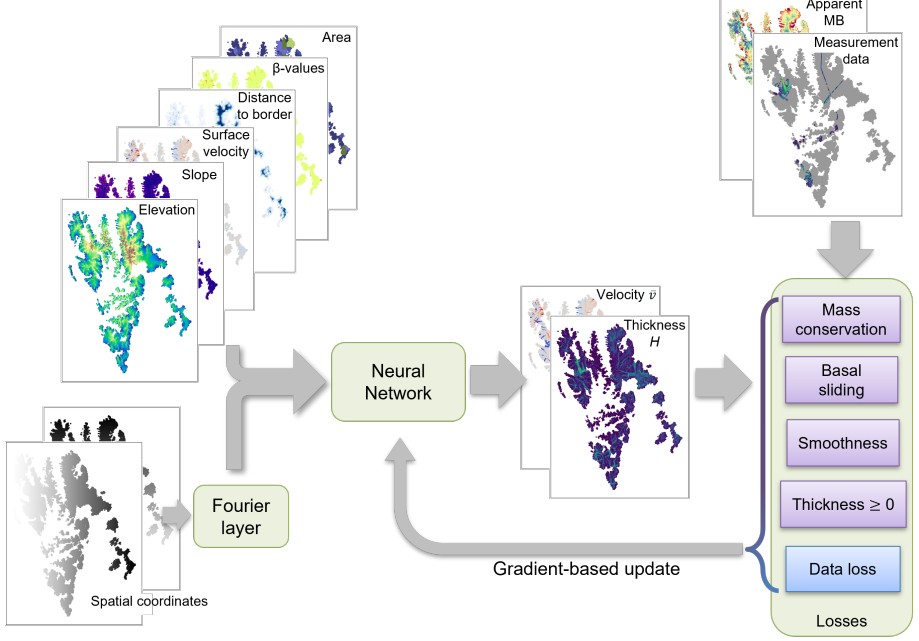

**Figure 1.** Physics-informed model with inputs, outputs, and components adding to the loss function. The physics-aware losses are in purple boxes. The Data loss in the blue box is the only loss depending on ice thickness measurement data. Surface velocity and $\beta$-values also add to the physics-aware losses. The connection is not shown to increase readability.

$$\mathcal{L}_{vel} = \begin{cases} (v_{\mathrm{s}} - \bar{v})^2 & \text{if } \bar{v} > v_{\mathrm{s}} \\ (v_{\mathrm{s}}(l_{\mathrm{lower}} + (1 - l_{\mathrm{lower}})\beta) - \bar{v})^2 & \text{if } \bar{v} < v_{\mathrm{s}}(l_{\mathrm{lower}} + (1 - l_{\mathrm{lower}})\beta) \\ 0 & \text{else} \end{cases} \quad \begin{array}{l} \text{with } \bar{v} \in \{\bar{v}_x, \bar{v}_y, \bar{v}_{\mathrm{mag}}\} \\ \text{and } \beta \in \{\beta_x, \beta_y, \beta_{\mathrm{mag}}\} \end{array}$$

(6)

$\mathcal{L}_{vel}$ is calculated separately for the x- and y-component and the magnitude of the depth-averaged velocity. As basal drag is most likely not the only drag the ice experiences, we decided to fix the lower bound as $l_{\mathrm{lower}} = 0.7$ in order to give more flexibility in the estimate.

We include two more physics-aware constraints to improve the model performance: First, the ice thickness is assumed to be smooth, so the model will penalize large spatial derivatives within the ice thickness prediction:

$$\mathcal{L}_{smooth} = (\nabla H)^2 \tag{7}$$

Secondly, ice thickness cannot be smaller than 0. Therefore, we add a loss component that punishes negative ice thicknesses to the loss function.

$$\mathcal{L}_{H>0} = \begin{cases} H^2 & \text{if } H < 0 \\ 0 & \text{else} \end{cases} \tag{8}$$

The final loss component is the data loss. It penalizes the deviation from the in situ ice thickness measurements and acts as the internal condition to solve the mass-conserving PDE. Each loss component will have a different scale, so we balance them with individual weights $\lambda_i$. Summing up all the loss components, we get the complete loss function as:

$$\mathcal{L} = \lambda_{mc}\mathcal{L}_{mc} + \lambda_{vel}\mathcal{L}_{vel} + \lambda_{smooth}\mathcal{L}_{smooth} + \lambda_{H>0}\mathcal{L}_{H>0} + \lambda_{data}\mathcal{L}_{data} \tag{9}$$

All the physics-aware losses are evaluated at any point in the study region. In contrast, the data loss is only evaluated wherever ice thickness measurements are available. We refer to the points with ice thickness measurements as *labelled*, and points without ice thickness measurements are referred to as *unlabelled*.

The training data is scaled to have a mean of 0 and a variance of 1. Before computing the physics-aware loss components, we scale the quantities back to their original units for physical consistency.

We tested a slim version of the PINN model on a one-dimensional data set of a single glacier. The results are given in App. C. The experiment shows the added value of introducing physics-aware loss components.

## 2.4 Validation

We evaluate the performance of the PINN model by calculating the root mean squared difference (RMSD) and the mean absolute percentage difference (MAPD) between the model prediction and the in situ ice thickness measurements. However, in situ measurements within a glacier are highly correlated due to their proximity. Therefore, a simple random split of the data into training and test datasets will not yield a realistic view of the model performance.

We employ a glacier-wise cross-validation (CV) approach as done by Bolibar et al. (2020) to better judge the model performance. This also allows us to make assumptions on how well the model will perform on a glacier without any measured ice thicknesses.

For the Leave-One-Glacier-Out (LOGO) CV, we chose seven glaciers that serve as test glaciers. In an alternating way, we train the model without one of those glaciers' ice thickness measurements.

It is important to note that only data labelled with ice thickness measurements of the test glacier is left out of the dataset. All the data needed to enforce the physical consistency for the test glacier stays in the training dataset. The mass-conserving PDE of Eq. (1) will still be solved at the test glacier but without enforcing internal conditions with ice thickness measurements.

Upon validation, the RMSD and the MAPD are calculated for the test glacier. All the test glaciers are thoroughly mapped with ice thickness measurements and differ in size, mean measured ice thickness, and location in Svalbard.

## 3  Data

In this study, we focus on the glaciers on the islands of Spitsbergen, Barentsøya, and Edgeøya. Glaciers in an active surge phase during the data acquisition period for the surface velocity are not considered. The information on active surge phases is collected from Koch et al. (2023).

### 3.1  Data management

We processed all the data needed for the training of the PINN using the open global glacier model (OGGM) framework developed by Maussion et al. (2019). OGGM is an open-source framework to simulate glacier evolution. It provides models for mass balance, distributed ice thickness, and ice flow, as well as downloading tools for glacier outlines, digital elevation models (DEM), and climate data. The mass balance model is a temperature index melt model relying on climate data.

OGGM saves all the information for each glacier separately in Glacier Directories. The Randolph Glacier Inventory (RGI), Version 6.0 contains the outlines for the glaciers (RGI Consortium, 2017).

Using the outlines, OGGM defines a spatial grid for each glacier. The grid resolution is adapted individually according to the size of the glacier. In our study region, the grid resolution ranges from $12\,\mathrm{m}$ to $200\,\mathrm{m}$. OGGM reprojects and scales the data for each glacier to the glacier grids. We collect these data and transform the coordinates from the individual grids into a common projection.

### 3.2  Surface Velocity Data

Millan et al. (2022) derived the surface flow velocity of the world's glaciers using image pairs acquired between 2017 and 2018 by Landsat 8, Sentinel-2, and Sentinel-1. They tracked glacier motion using a cross-correlation approach. The resolution of the velocity product is $50\,\mathrm{m}$ with an estimated accuracy of about $10\,\mathrm{yr\,m^{-1}}$. Using OGGM, velocity in x- and y-direction and velocity magnitude are projected onto the individual glacier grids and then smoothed with a two-dimensional Gaussian filter.

We introduce the aforementioned $\beta$ value (see Sect. 2.2) to incorporate the influence of basal sliding on the measured surface velocity $\boldsymbol{v}_{\mathrm{b}} = \boldsymbol{\beta v}_{\mathrm{s}}$. Following Millan et al. (2022), we set $\beta$ equal to 0.1 in areas where the ratio between slope and observed surface velocity is greater than $0.001\,\mathrm{yr\,m^{-1}}$ and modulate up to 0.9 for areas where the ratio is less than $0.001\,\mathrm{yr\,m^{-1}}$. For each point, we compute three $\beta$ values from the surface velocities in the x- and y-direction and the magnitude of the surface velocity.

### 3.3  Apparent mass balance

The apparent mass balance is the difference between the point-wise mass balance and the thickness change rate $dh/dt$ at each grid point. The mass balance at each point of a glacier grid is estimated using the *ConstantMassBalance* model from OGGM. It calculates the average mass balance during a chosen period from given climate data, calibrated with geodetic mass balance data from Hugonnet et al. (2021). To match the acquisition period of the surface velocity, we set the climate period for the mass balance model to 2016-2018.

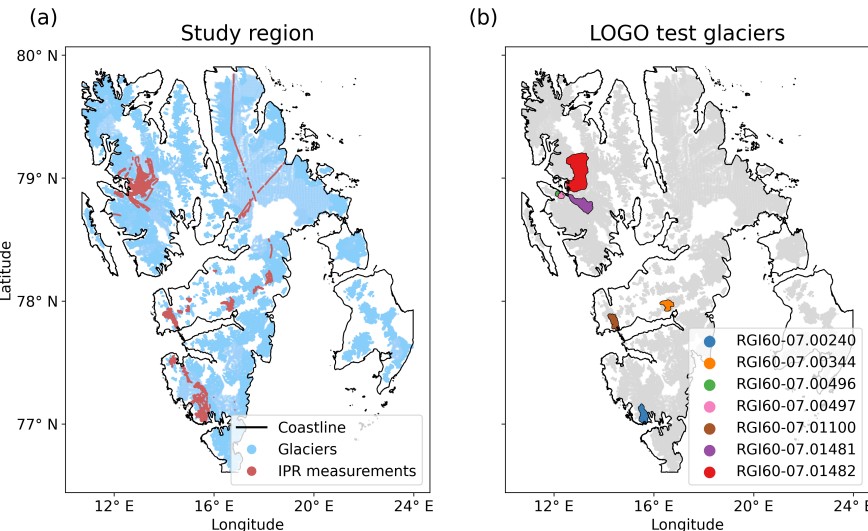

**Figure 2.** Glaciers in the training dataset. (a) The locations of in situ measurements are marked in red. (b) The locations of the test glaciers. The coastline is retrieved from Moholdt, G. et al. (2021).

The rate of thickness change $dh/dt$ is retrieved from ASTER DEM differences between 2015 and 2019 (Hugonnet et al., 2021). The data is projected onto the glacier grids using OGGM and then smoothed with a two-dimensional Gaussian filter.

### 3.4 Thickness Data

The data-driven machine learning model needs ice thickness data as ground truth for its supervised training. The Glacier Thickness Database (GlaThiDa) is a comprehensive public database of in situ glacier thickness measurements collected from various studies (GlaThiDa Consortium, 2020). Version 3.1.0 was released in 2020 with nearly one million measurements from ice penetrating radar (IPR) on 207 glaciers or ice caps in Svalbard.

   In situ ice thickness measurements are not error-free. GlaThiDa lists reported uncertainties of almost 80% of the mea-
195 surements in Svalbard. The mean and standard deviation of the thickness uncertainty are 6.2 m and 4.4 m with a maximum uncertainty of 21 m.

   During the preprocessing, the measurements are projected onto the OGGM glacier grids by aggregating and averaging them at their nearest point on the glacier grid. We only consider aggregated ice thicknesses where the average acquisition year is after 2000. That leaves us with 27 554 points labelled with ice thickness on 65 glaciers.

### 3.5 Auxiliary data

Adding to the data that we need to impose the physics-aware losses, we also feed the network with extra information from auxiliary data as input features. We chose the features because they were easily available through OGGM and are related to the glacier's ice thicknesses. In Appendix F we analyze how each of the features impact the model output. The elevation of each

| RGI ID | Area [km$^2$] | IPR Mean [m] | IPR StD [m] | Survey year Mean | Num of Measurements |
|---|---|---|---|---|---|
| RGI60-07.00240 | 64.211 | 216.5 | 94.3 | 2008 | 1200 |
| RGI60-07.00344 | 36.087 | 161.3 | 70.1 | 2002 | 667 |
| RGI60-07.00496 | 5.016 | 82.1 | 39.5 | 2010 | 1069 |
| RGI60-07.00497 | 6.249 | 87.6 | 43.8 | 2010 | 884 |
| RGI60-07.01100 | 50.408 | 146.3 | 61.1 | 2012 | 1684 |
| RGI60-07.01481 | 108.297 | 240.6 | 97.1 | 2015 | 695 |
| RGI60-07.01482 | 378.765 | 317.9 | 171.8 | 2015 | 2202 |

**Table 1.** Area of each test glacier together with the mean and standard deviation of the IPR ice thickness measurements, mean acquisition year, and number of IPR measurements.

point comes from the global DEM from Copernicus DEM GLO-90, which was acquired from 2010 to 2015 (Copernicus). The slope is then computed by OGGM based on the glacier's smoothed topography and over the length of a grid cell on the glacier. The distance to the border of the glacier outline is computed for each point within a glacier. The glacier area is also retrieved from the RGI.

The full dataset of points with and without ice thickness labels consists of over 3 million data points from the grids of 1465 glaciers. Figure 2 (a) displays the considered glaciers in light blue and the acquisition lines of the IPR measurements in red.

## 3.6   Test glaciers

Seven glaciers with the most in situ measurements are chosen as test glaciers for the LOGO CV. They differ in size and mean thickness and are located in different areas of Spitsbergen. No glaciers on Barentsøya and Edgeøya are mapped well enough to use them as test glaciers.

Figure 2 (b) shows the location of the test glaciers. Table 1 lists the area, mean and standard deviation of the measured ice thicknesses, and mean of the survey year for those glaciers. Measurements on glaciers RGI60-07.00496 and RGI60-07.00497 are all from one survey, while the others are from multiple surveys carried out in different years.

## 4   Results

The LOGO CV produces seven models with the same architecture but different model weights. Each model was trained on all the unlabelled data to enforce the physical constraints at every point. After putting aside the test glacier's labelled data, the remaining glaciers' labelled data was randomly split into 60% training and 40% validation data.

The in-sample performance is measured based on the validation data the model did not see during the training. Table 2 lists the RMSD and MAPD for the in-sample validation data. The PINNs predict glacier ice thickness with a mean in-sample RMSD of 30 m corresponding to a MAPD of 36%.

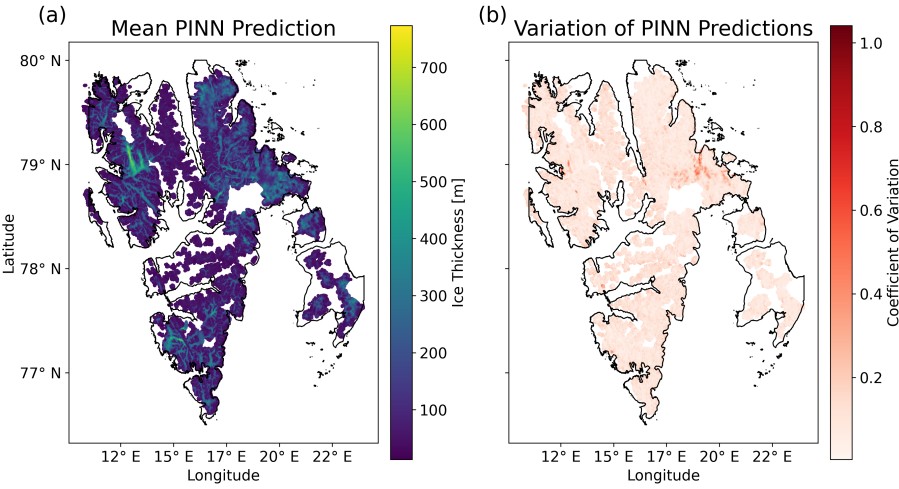

**Figure 3.** Mean (left) and coefficient of variation (right) of the ice thickness predictions from all seven models from CV.

The training and validation data are spatially correlated. Therefore, the in-sample evaluation of the model probably overestimates its performance.

Figure 3 shows the mean ice thickness prediction and the coefficient of variation over all seven LOGO models for the study region. The coefficient of variation measures the variability in relation to the mean ice thickness at each point of the grid. 90% of the points have a variability below 0.16. As the in-sample validation scores of each model are also similar, we are confident that the method is robust to varying labelled data. The PINN models agree with their predictions, although they were trained with different sets of ice thickness measurements as target data.

## 4.1 LOGO results

The model performance for the test glaciers delivers insights on the performance we can expect for glaciers where we do not have any in situ measurements. Table 2 shows the results of the LOGO validation for each of the test glaciers. As expected, the RMSD and MAPD are significantly higher than for the in-sample validation data. Figure 4 shows the difference between the model's ice thickness estimate and the in situ measurements along the IPR acquisition lines for the seven LOGO test glaciers that were excluded from the dataset during training. Overall, the models underestimated the ice thickness. The ice thickness estimates for the entire grids of the glaciers are displayed in Fig. D1.

The test glaciers differ significantly in mean ice thicknesses (see Table 1). For thinner glaciers like RGI60-07.00496 and RGI60-07.00497, the MAPD is very high, although their RMSD is comparable to the in-sample scores. The RMSD of glacier RGI60-07.01482 is four times as high as the mean RMSD score of the in-sample glaciers, but as, on average, in situ measurements are very thick, the MAPD is closer to the in-sample MAPD than the MAPD for an on average thinner test glacier. This makes it clear that considering both validation scores is necessary to view the model performance accurately. Another exam-

| Test glacier | In-sample validation | | LOGO validation | |
| :---: | :---: | :---: | :---: | :---: |
| ID | RMSD [m] | MAPD | RMSD [m] | MAPD |
| RGI60-07.00240 | 30 | 0.36 | 77 | 0.30 |
| RGI60-07.00344 | 31 | 0.38 | 55 | 0.40 |
| RGI60-07.00496 | 33 | 0.35 | 38 | 0.49 |
| RGI60-07.00497 | 30 | 0.36 | 46 | 0.94 |
| RGI60-07.01100 | 31 | 0.34 | 40 | 0.65 |
| RGI60-07.01481 | 30 | 0.36 | 83 | 0.77 |
| RGI60-07.01482 | 29 | 0.34 | 124 | 0.54 |
| **Mean** | 30 | 0.36 | 66 | 0.58 |

**Table 2.** Results of the LOGO CV. The in-sample validation scores are measured from the validation set that contains in situ data from every glacier but the test glacier that was left out during training. The LOGO validation scores are measured from the in situ data for the left out test glacier.

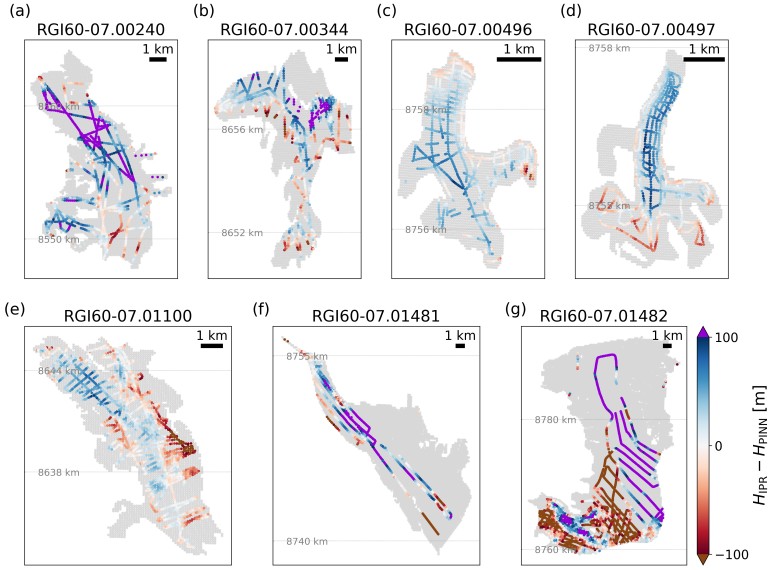

**Figure 4.** Difference between the predicted ice thickness and the IPR ice thickness measurements for the seven LOGO test glaciers.

ple of that would be the comparison of performances on glaciers RGI60-07.00240 and RGI60-07.01481. They have similar measured ice thicknesses and RMSD scores but their MAPDs differ greatly.

Over all seven test glaciers, the mean RMSD and the mean MAPD are about 66 m and 0.58%, respectively, i.e. significantly worse than the in-sample metric. This indicates that the PINN is overfitting on glaciers with thickness measurements.

## 4.2 Comparison to other estimates

As we do not have full coverage with in situ measurements, the model scores only represent the model's performance at the acquisition lines of the IPR measurements. Therefore, we compare the ice thickness predictions to the estimates of Millan et al.
(2022), Farinotti et al. (2019), and van Pelt and Frank (2024) to see how much the estimates differ. All of those ice-thickness products are derived from physics-based models. Farinotti et al. (2019) estimated ice thickness using an ensemble of up to five models; therefore, the name *consensus* estimate. Millan et al. (2022) rely on a single model that uses the shallow-ice approximation and surface motion to compute ice thickness. van Pelt and Frank (2024) use two inverse methods, one for small and one for larger and surging glaciers, to create their ice thickness product.

The plots in Fig. 5 show the scatter plot of the ice thicknesses of the PINN ensemble estimates versus the three other estimates for each point in the study region. The solid red line shows the linear fit between the two ice thickness products, while the black dashed line is the 1:1 line. The values of slope and intersect for the linear fits indicate that the PINN estimates agree slightly less on the ice thickness at each grid point than the three physics-based models. Comparing the mean ice thickness estimate from the ensemble of PINNs to the estimates of physics-based models shows that the deviations from the other estimates are
within the range of the differences between the physics-based models. Mean absolute differences (MAD) between the PINN and physics-based predictions are all close, with a mean of 34 m. The MAD between the physics-based models is in the same order of magnitude, and their mean MAD is at 34.3 m.

    Since physics-based models also work with simplifications of ice dynamics, their ice thickness products can not be taken as definitive truth. Comparing the predictions of the PINN ensemble to those only serves to estimate the qualitative validity of the
ice thickness predicted by the PINN ensemble. The overall correlation between the ice thickness estimates leads us to believe that the PINN ensemble produces valid ice thickness estimates.

## 4.3 Depth-averaged velocity

The models estimate the depth-averaged velocities in x- and y-direction. There is no ground truth data for the depth-averaged velocities, so we can not evaluate the models's accuracy. However, during training, the loss of the predicted velocities is reduced
significantly, showing that the constraints of Eq. (3) are enforced.

## 5 Discussion

Evaluating the PINN performance with the in-sample validation set and comparing predicted ice thicknesses to other products suggests that the PINN produces reasonable ice thickness estimates. However, testing the model with a glacier-wise CV scheme unveils the lack of generalizability to glaciers without any measurement data. The differences between the predicted
ice thicknesses and the measurements are much higher for the test glaciers of the LOGO CV than for the in-sample validation.

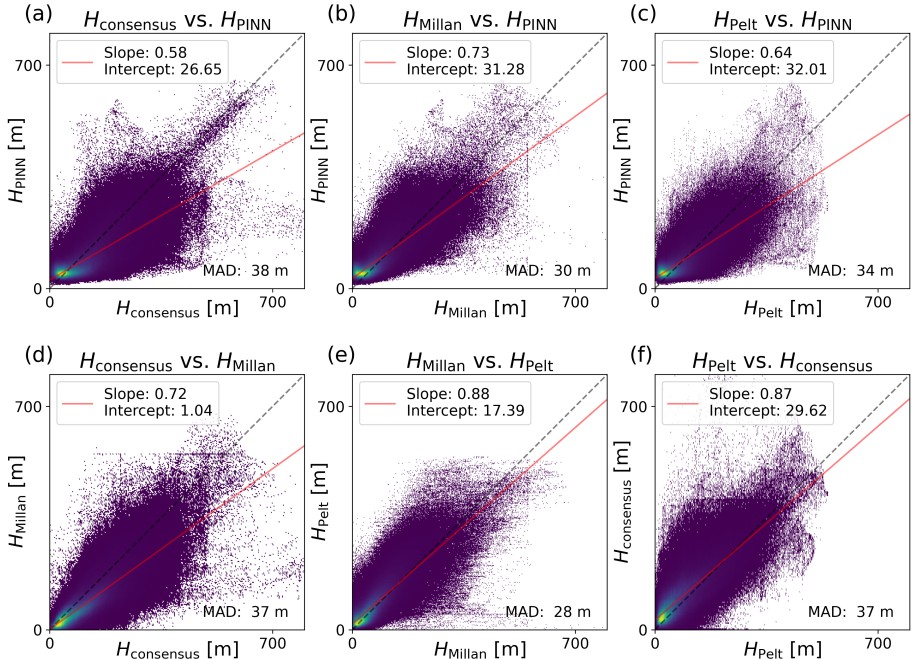

**Figure 5.** Comparisons of ice thickness estimates: a) Ice thickness prediction from the ensemble of PINNs versus ice thickness estimates of Millan et al. (2022), Farinotti et al. (2019), and van Pelt and Frank (2024). b) Ice thickness estimates from physics-based models against each other. The color indicates the point density (the brighter the denser). The solid red line shows the linear fit between the two ice thickness products. The black dashed line is the 1:1 line.

We identified several factors that may improve generalizability but are also challenging to address. The schematic of Fig. 6 shows an overview of the domains and the particular issues we judge as the most pressing to address.

## 5.1 Data

Although PINNs are generally relatively less dependent on training data than purely data-driven methods, their performance relies on the quality of input data (Iwasaki and Lai, 2023). This study collected the thickness measurements from 65 glaciers. The individual measurements lie close together along the acquisition lines. As a result, most of the measurements have high spatial correlations with each other. The amount of independent training data to learn the physics of glaciers is, therefore, far less than the actual number of measurements. On the other hand, redundant data introduces a bias. To reduce the overfitting, we could reduce the correlations in the training dataset by averaging or sub-sampling the observations, for example. This should improve the performance on glaciers without any labelled training data.

Secondly, the training data is not aligned temporally. In situ ice thickness measurements were collected between 2000 and 2017, while the surface velocity was derived from satellite data acquired between 2017 and 2018. Using surface velocity for the same year as ice thickness was measured would result in a better estimate of the ice flux for the labelled data and therefore

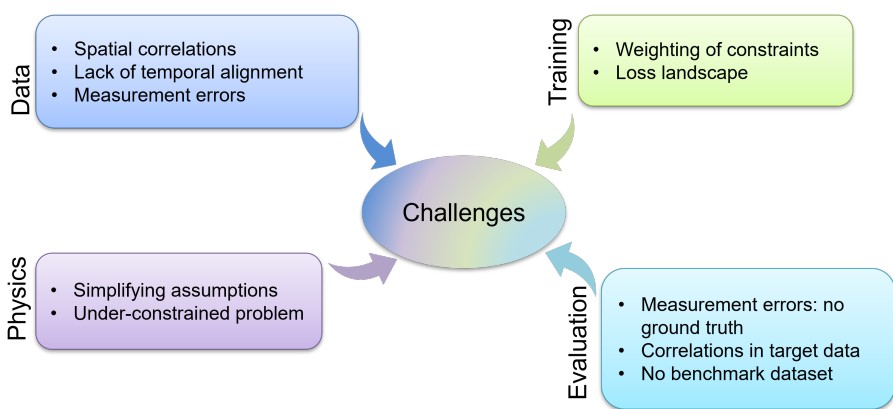

**Figure 6.** Challenges for PINNs in a real-world setting like the prediction of glacier ice thickness. The separate realms are interfering with each other, complicating the optimization of the model. Weighting of the physical constraints could have the biggest positive benefit.

improve physical consistency within the model. However, this is only important if either ice thickness or velocity changes significantly with time.

Lastly, we do not consider measurement errors in our dataset. Ice thickness measurements from ice-penetrating radar, for example, are subject to errors due to varying density of glacier ice but also due to unknown thickness of snow and firn layers (Lindbäck et al., 2018). Future work should account for measurement errors with standard uncertainty quantification methods or even introduce error margins to the loss components as Morlighem et al. (2011) did.

## 5.2 Model training

Training PINNs is difficult (Xu et al., 2023). One major challenge is to find the optimal balance between the multi-scale contributions of each loss component, which is also reported in other studies using multiple loss components in their PINNs (Iwasaki and Lai, 2023; Cheng et al., 2024). We empirically set the loss weights to a fixed value for the entire training process.

In an experiment to test the importance of the loss components, we found that the relative importance is not very pronounced (see Appendix E). Therefore, we assume that the individual loss components are not optimally weighted in the reported model.

A more sophisticated approach would be to automatically update the weights of the loss components during the training. This introduces some computational overhead but prevents the model from minimizing certain loss components faster than others (Wang et al., 2023).

The multiple loss components create another challenge: the loss landscape is highly complex, and finding its global minimum is difficult. Recently, Rathore et al. (2024) investigated different optimizers for PINNs and showed that combined first- and second-order optimizers lead to faster convergence. Implementing their newly introduced second-order optimizer could

improve the PINN convergence. Development of new optimization strategies is a rapidly evolving area of research, and we expect significant advances to be achieved soon.

## 5.3 Physical constraints

Our model uses physics-aware loss components to enforce physical consistency. However, the physics-aware losses are based on simplifications to make the problem tractable. We identified several challenges that are tied to the incorporation of physical constraints.

Firstly, we use a simple model from OGGM out of the box without further calibration to derive the mass balance at every

315 point in the study region. The mass balance data appears in the computation of the mass conservation loss, but if it is erroneous, the model will never be able achieve perfect mass conservation. Therefore, using a more sophisticated mass balance model and calibrating it for our purpose could enhance physical consistency.

Another option could be to use in situ mass balance data. This way, we circumvent the need for a mass balance model. The mass conservation loss would only be evaluated where data is available. This, however, would come with two restrictions. First,

we would not be able to train the model in the entire study region. Secondly, also in situ mass balance data is not error-free. We would have to make a careful selection of the data to not introduce even more uncertainty.

A different way to improve the physical consistency is through a better estimate of the depth-averaged velocity. In the current model the estimate of depth-averaged velocity is coupled to estimating the amount of basal sliding and using a parameter as a

325 lower bound for the vertical integration of the velocity. For now, the estimate of the sliding velocity is based on a simplified approach using a threshold calibrated by Millan et al. (2022). We could, in principle, circumvent the need for this parameter by using surface velocity data acquired during the winter months, when basal sliding is inhibited. The measurable surface velocity will have less contribution from basal sliding and we could avoid estimating the $\beta$-parameters. This would eliminate one source of uncertainty.

We also want to mention that there are several processes affecting ice dynamics, especially in Svalbard, that are not very simplified or neglected in the model. One example is that our model assumes ice to be incompressible, when Svalbard glaciers actually have thick firn layers (Pälli et al., 2003). The varying density could introduce a non-negligible densification term in the mass balance Eq. (1).

Another example is the assumption of a temperature-independent creep coefficient A. Many glaciers in Svalbard are believed

to be polythermal (Glasser, 2011). So, the creep coefficient may vary within the ice, affecting the validity of our lower boundary for the estimate of depth-averaged velocity. However, the influence of these effects should carefully be weighed against the possibility of introducing errors if we decide to include better representations of these processes.

Lastly, one major challenge is that we deal with a highly under-constrained problem. We only provide the model with the ice thickness measurements as a sort of internal condition, but we do not provide boundary conditions. Also, the depth-averaged ve-

340 locity is only loosely constrained by a set of inequalities. Therefore, it would be beneficial to incorporate additional constraints

like momentum conservation to actually derive depth-averaged velocity instead of estimating it. While this is technically easy to do, it comes at the cost of introducing uncertainties from approximating required parameters. To enforce momentum conservation, we would to need make assumptions about ice viscosity and resistance from the bedrock, for example. Depending on the data quality, we risk introducing more uncertainty instead of improving the physical consistency.

In our view, the two elements that are most promising to improve the model performance, if revised, are the modelling of mass balance for Svalbard and the choice of surface velocity data for the estimation of depth-averaged velocities.

## 5.4 Evaluation

In geospatial machine learning, evaluation is generally challenging (Rolf, 2023). As mentioned in Sect. 2.4, the in situ measurements are heavily correlated since they are clustered on only a fraction of the glaciers. Therefore, we employ a spatially-aware evaluation method to estimate the true model performance. However, the CV procedure only includes seven of 65 glaciers with measurements. Moreover, we have varying numbers of IPR measurements for the evaluation of each of the test glaciers as already mentioned in Sec. 4.1. There is no guarantee that we fully capture the model performance with our approach.

Additionally, in situ ice thicknesses are subject to measurement errors, and some measurements might have higher errors than others. To be as precise as possible when evaluating the model performance, we should consider the trustworthiness of every ice thickness label.

Ultimately, our problem has no benchmark dataset, so it is impossible to know the model performance exactly. Although we compare our ice thickness estimates with others, these also have errors that are not well constrained and are, in no respect, benchmarks that can be used for uncertainty quantification. It is, therefore, difficult to state which method considered here produces the most reliable ice thickness estimate.

Despite the above-mentioned limitations, we show that a relatively simple PINN can produce reasonable ice thickness estimates while treating an entire area and not only a single glacier at once. Although the lack of high quality data is an overarching challenge that can hardly be overcome, we expect that by implementing the proposed adjustments in data curation, model training, and physical constraints, the physical consistency and accuracy of the model will be improved. Without changing the dataset we believe that optimizing the loss weights $\lambda$ would have the biggest positive benefit, as the optimal configuration depends on the noise in the data Iwasaki and Lai (2023). This may especially be the case for glaciers without measured ice thicknesses.

## 6 Conclusions

We have demonstrated that it is possible to train a physics-aware machine learning model to produce ice thickness estimates for multiple glaciers, including glaciers without any ground truth ice thickness: in other words, out of sample targets. We deploy a relatively simple physical constraint by imposing mass conservation in the loss function of the PINN. This serves as a proof

of concept that physics-informed models can not only be applied to one single closed system but, together with auxiliary data, can make meaningful predictions for entire regions.

More complex approaches and physical constraints could be employed (Karniadakis et al., 2021) and would, we anticipate, improve the results further. Nonetheless, we demonstrate that physics-aware machine learning is a promising approach for tackling this geophysical problem where a physical law and multiple conditions provide constraints for the solution. There are many other geophysical problems where, for example, including conservation of mass, energy, or momentum would provide a similarly effective constraint and would lead to a more scientifically meaningful result, as breaching such constraints is non-physical.

*Code and data availability.* The code and data that was used to train and evaluate the model as well as generating the figures are available at https://doi.org/10.5281/zenodo.13834016. Additionally, the code can be viewed at https://github.com/viola1593/glacier_pinn.

## Appendix A: Relation between surface and depth-averaged velocity to set $l_{\text{lower}}$

To derive a relation between the surface velocity and the depth-averaged velocity, we follow the analysis in Cuffey and Paterson (2010). Let $u$ be the x-component of velocity and $H$ be the ice thickness. Assuming parallel flow, the glacier deforms in simple shear, so the only nonzero deviatoric stress is $\tau_{xz}$, and the z-component of the velocity is also zero. Therefore, the creep relation derived by Glen (1955) simplifies to

$$\frac{1}{2}\frac{du}{dz} = A\tau_{xz}^n \tag{A1}$$

where $A$ is the creep parameter and $n$ the creep exponent. $A$ is, in general, dependent on the temperature of the ice, so $A = A(T)$. We assume a linear increase of shear stress along the glacier depth

$$\tau_{xz} = \tau_b\left[1 - \frac{z}{H}\right] \tag{A2}$$

with $\tau_b$ being the shear stress at the bed of the glacier. We further assume constant temperature within the ice so $A$ does not depend on z. If we integrate A1 along the vertical direction up to $z$ we get

$$u(z) = u_b + \frac{2A}{n+1}\tau_b^n H\left[1 - \left[1 - \frac{z}{H}\right]^{n+1}\right]. \tag{A3}$$

Accordingly, the velocity at the surface is given by

$$u_s = u_b + \frac{2A}{n+1}\tau_b^n H \tag{A4}$$

and integrating A3 along the vertical axis to derive the depth-averaged velocity we get

$$\bar{u} = u_b + \frac{2A}{n+2}\tau_b^n H. \tag{A5}$$

A is assumed to be constant although it depends on temperature and other variables that change within a glacier profile. Temperate glaciers are nearly isothermal, whereas in cold-based glaciers, the temperature increases with a smaller distance to the bed. The highest values of A are found near the glacier bed. Therefore, the shear deformation is concentrated closer to the base than in a temperate glacier. The velocities within the bottom half of the glacier are sensitive to the value of A as it multiplies stress to the power of n. As stress decreases in the upper half of the glacier, the velocity is insensitive to the values of A there.

From A4 and A5, we can derive the relation between the surface velocity and the depth-averaged velocity in the case of parallel flow and if there is no basal sliding:

$$\frac{\bar{u}}{u} = \frac{n+1}{n+2} = 0.8 \tag{A6}$$

for $n = 3$. After all, parallel flow in a glacier is a strong assumption, and $n = 3$ is not always the case. Therefore, we set $l_{\text{lower}}$ to 0.7 to allow more flexibility in estimating the depth-averaged velocity.

## Appendix B: PINN architecture and training

The PINN employed in this work consists of a fully-connected neural network with 8 layers and 256 neurons each. We chose Softplus as an activation function after each layer as it is infinitely differentiable.

$$\text{Softplus:} f(x) = \log(1 + \exp(x)) \tag{B1}$$

The Fourier feature encoding layer maps each of the spatial coordinates to a 32-dimensional vector using a matrix B with entries drawn from a Gaussian distribution with a variance of 10.0. The loss weights $\lambda_i$ are set to keep all the loss components roughly in the same order of magnitude. We chose the Adam optimizer with default settings from PyTorch and a learning rate of 0.0001 at a batch size of 8192. In the LOGO cross-validation, each model is trained for 100 epochs.

## Appendix C: Experiment on synthetic 1D data

As an illustrative example to better explore the validity and performance of the PINN model, we created a synthetic dataset of a one-dimensional glacier. We generated the synthetic data using an artificial bedrock and ice thickness from where we calculated basal stress $\tau_b = \rho * g * H * \alpha$ (van der Veen, 2013) with $\rho$ being the density of ice, $g$ the gravitational acceleration and $\alpha$ the surface slope.

The velocity is given by Eq. A4 where we choose $A = 2.4 * 10^{-24}$, $n = 3$, and $u_b = 0$. The apparent mass balance is calculated from mass conservation Eq. 1. During the training, we provided the model with the x values, surface velocity, and the beta value as described in 3.2. The model only received 10 points labelled with ice thickness unevenly spread over a domain of 5 km length. We did not include the auxiliary data like the point's distance to the glacier border or the area of the glacier a point lies on, as this does not make sense in the one-dimensional case. For this test case, we switched off the Fourier feature encoding layer in the model, as well as the loss on the magnitude of the depth-averaged velocity prediction.

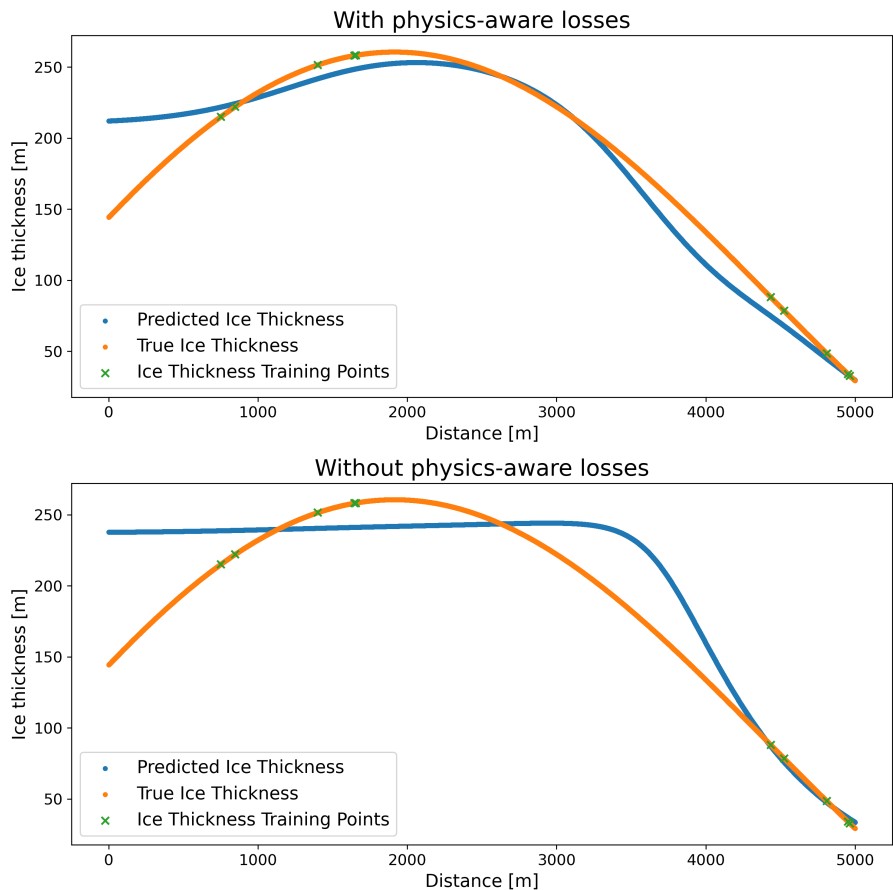

**Figure C1.** Ice thickness predictions from the model with and without physical constraints for the synthetic test case.

Figure C1 compares the results of the ice thickness prediction from the model with physics-aware losses and without. Clearly,
in the regime where we did not provide data for the ice thickness. The model profits from the physical constraints that keep the prediction closer to the true ice thickness than the prediction from a model without physics-aware losses.

**Appendix D: PINN ice thickness prediction on LOGO test glaciers**

The ice thicknesses of the seven test glaciers were estimated by the model that was trained without the in situ measurements of the respective glaciers as ground truth data. The results are displayed in Fig. D1

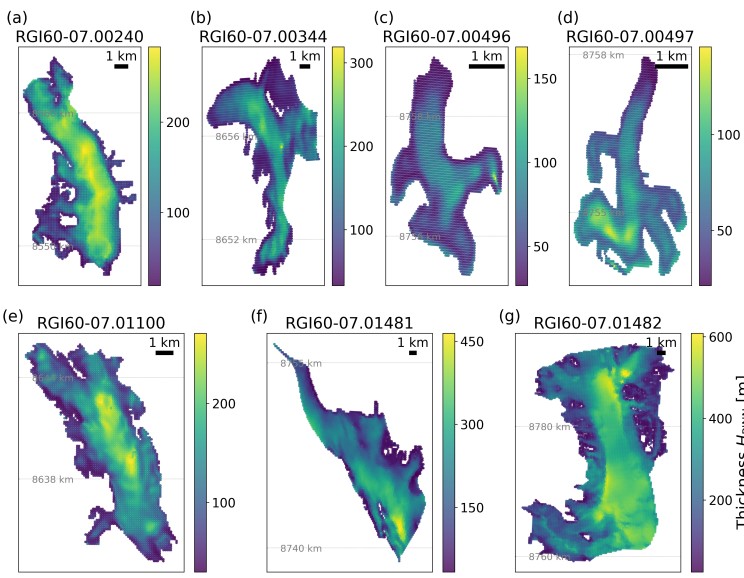

**Figure D1.** Ice thickness prediction for the seven LOGO test glaciers.

## Appendix E: Importance of physics-aware loss components

The importance of the individual loss components is tricky to evaluate, as we are dealing with unevenly distributed, correlated, noisy in situ measurements as labels to evaluate the PINN performance. Despite this, we ran the LOGO experiments in which we set the weight of each of the loss components to zero one after another. We then compared the performance to the scores of the models reported in Sec. 4.1 by calculating a relative RMSD on the validation set as $\mathrm{RMSD}_{\mathrm{rel}} = (\mathrm{RMSD}_{\mathrm{reported}} - \mathrm{RMSD})/\mathrm{RMSD}_{\mathrm{reported}}$. The relative RMSD will be positive if the score improves and negative if the score gets worse by setting the weight of a loss component to 0.

The relative differences are below 5% on average. It is interesting to see that, while the scores on the in-sample validation data improve on average when switching off the physics-aware loss components (Table E1), the scores on the out-of-sample test glaciers get worse on average (see Table E2). This fits with our intuition that the model is overfitting on the in situ ice thickness data that we provide it with during training. The physics-aware loss components act like a regularization while demanding physical consistency.

Nevertheless, we want to emphasize again that the configuration of the loss weights is certainly not optimal, as we discussed in Sec. 5.2, so there might be another distribution of importance if all the loss components are better balanced. Also, as already mentioned, this experiment depends a lot on the dataset, so the importance of loss components also only applies to this specific study.

| Test glacier RGI ID | Mass conservation loss $RMSD_{rel}$ | Velocity loss $RMSD_{rel}$ | Smoothness loss $RMSD_{rel}$ | Negative thickness loss $RMSD_{rel}$ |
|---|---|---|---|---|
| RGI60-07.00240 | -0.033 | 0.000 | -0.033 | -0.033 |
| RGI60-07.00344 | 0.032 | 0.032 | 0.065 | 0.000 |
| RGI60-07.00496 | 0.000 | 0.030 | 0.061 | 0.000 |
| RGI60-07.00497 | 0.033 | 0.000 | 0.033 | 0.000 |
| RGI60-07.01100 | 0.032 | 0.032 | 0.032 | 0.000 |
| RGI60-07.01481 | 0.000 | 0.033 | 0.033 | -0.033 |
| RGI60-07.01482 | 0.034 | 0.034 | 0.034 | 0.000 |
| **Mean** | **0.014** | **0.023** | **0.032** | **-0.010** |

**Table E1.** Relative RMSD scores for in-sample validation for each LOGO CV model.

| Test glacier RGI ID | Mass conservation loss $RMSD_{rel}$ | Velocity loss $RMSD_{rel}$ | Smoothness loss $RMSD_{rel}$ | Negative thickness loss $RMSD_{rel}$ |
|---|---|---|---|---|
| RGI60-07.00240 | -0.091 | -0.195 | -0.013 | -0.013 |
| RGI60-07.00344 | -0.018 | -0.036 | 0.000 | 0.018 |
| RGI60-07.00496 | -0.132 | -0.184 | 0.105 | -0.053 |
| RGI60-07.00497 | 0.000 | -0.087 | 0.000 | -0.022 |
| RGI60-07.01100 | 0.025 | -0.125 | -0.150 | 0.000 |
| RGI60-07.01481 | 0.000 | 0.072 | -0.157 | -0.024 |
| RGI60-07.01482 | 0.008 | 0.032 | -0.048 | -0.048 |
| **Mean** | **-0.030** | **-0.075** | **-0.038** | **-0.020** |

**Table E2.** Relative RMSD scores for each LOGO CV test glacier.

## Appendix F: Importance of input features

The physics-aware model does not only take features that it would need to evaluate the physics-aware losses but also auxiliary data. To gain insights into the model's inner workings and evaluate how it handles the auxiliary data, we estimated the feature importance on the ice thickness predictions.

One way to approximate feature importance is by calculating Shapley values. This concept is rooted in game theory and estimates a player's contribution to a cooperative game (Shapley, 1953). Shapely values represent the contribution of each feature to the model prediction.

However, analytically deriving Shapley values for deep neural networks is very costly (Höhl et al., 2024). Therefore, Shapely values are approximated using techniques like the SHapley Additive exPlanations (SHAP) framework introduced by Lundberg and Lee (2017).

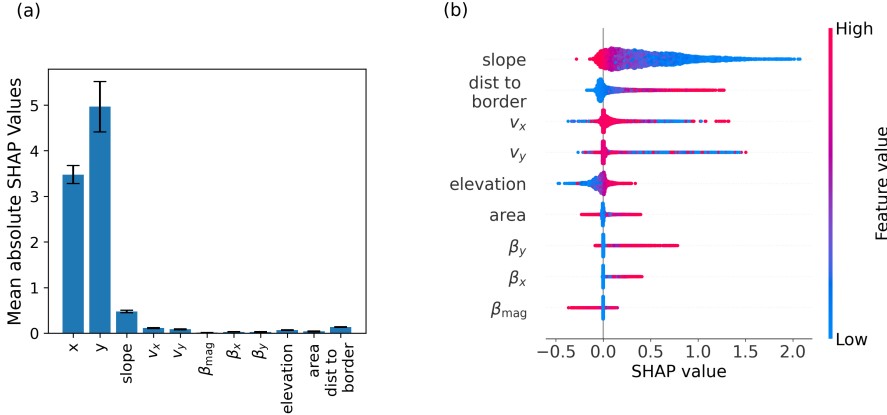

**Figure F1.** SHAP analysis: (a) Mean and standard deviation of the absolute SHAP values over all seven LOGO CV models. The values were first averaged for each model separately and then averaged for all seven models. (b) SHAP values for each datapoint by feature. The colour shows the relative value the feature takes for each datapoint. The SHAP values are calculated for the model trained without data from glacier RGI60-07.00240.

Within the framework, they describe a method with improved computational performance to estimate SHAP values for deep networks: Deep SHAP.

We used the implementation in the Captum library (*DeepLIFTShap*) (Kokhlikyan et al., 2020) to calculate SHAP values for our network. The validation data served as a representative subset of the entire dataset to save computational resources. We

calculated the SHAP values for each of the seven models from the LOGO CV.

The framework explains feature contributions to the model prediction, usually for purely data-driven machine learning. In Fig. F1 (a) high values signify a high impact on the output of the model; low values signify a low impact. For our PINN, the spatial coordinates are by far the most important features. This is expected as they define the domain in which we want to find a solution for the mass conservation PDE. Figure F1 (a) shows the mean absolute SHAP values for the features over all seven

models from the LOGO CV.

Besides the spatial coordinates the slope has the biggest impact on the prediction. Figure F1 (b) shows the impact of the features on the output of the model for each data point separately. For better readability, the plot shows the result of the SHAP analysis for only one of the models from the LOGO CV, as they are all similar.

The colour indicates the feature values: red signifies a relatively high value (within the range of the feature), and blue

signifies a relatively low value. For example, the plot in Fig. F1 (b) shows that high slope values lead to rather small values for the predicted ice thickness, while low values increase the predicted ice thickness. This is what we would expect given that ice thickness and slope are indirectly proportionally related in the SIA; steep slopes lead to thinner ice.

The SHAP values for the distance-to-border feature tell us that the model thinks that at the border the ice thickness should be smaller than within the glacier. For the surface velocity values the interpretation is less clear, also because we only see the

component-wise features. High surface velocities do not seem to have much impact on the ice thickness prediction, although, following glacier physics, they should have a strong influence on ice thickness.

We want to emphasize that the SHAP analysis has several limitations. First of all, it expects features to be independent of each other, which clearly is not the case here. The three $\beta$ values are derived from slope and velocity values for example. Also, the analysis depends very much on the dataset. SHAP tries to replicate the model behaviour, and the model is trained with our specific dataset. Therefore, the results can only show the impact of the features on the ice thickness prediction for our specific dataset and model setup. Additionally, machine learning models can only learn correlations from the data. Causal relationships can not be extracted. Hence, we can not derive universal feature importance from the analysis.

However, the results from the analysis are what we would expect from physical considerations. Therefore, it serves as a sanity check if the model is retrieving sensible correlations.

*Author contributions.* VS conceived the study, conducted the analysis and developed the ML framework and wrote the manuscript with contributions from all authors. JLB provided advice on how to incorporate the physics into the model and model training setup. XZ conceived the study and contributed to the interpretation of the results.

*Competing interests.* The authors declare that they have no conflict of interest.

*Acknowledgements.* This work is jointly supported by the German Federal Ministry for Economic Affairs and Climate Action in the framework of the "National Center of Excellence ML4Earth" (grant number: 50EE2201C), the German Federal Ministry of Education and Research (BMBF) in the framework of the international future AI lab "AI4EO – Artificial Intelligence for Earth Observation: Reasoning, Uncertainties, Ethics and Beyond" (grant number: 01DD20001), the Helmholtz Association under the joint research school "Munich School for Data Science - MUDS", and the Munich Center for Machine Learning (MCML). Jonathan Bamber also received funding from the European Union's Horizon 2020 research and innovation programme through the project Arctic PASSION (grant number: 101003472). We thank Dr. Ward van Pelt and Thomas Frank for providing us with their ice thickness dataset that was used for comparison in this study. We also want to thank Thomas O. Teisberg for the insightful suggestion on using Fourier feature embedding to improve the model performance. Lastly, we want to thank Adrian Höhl for sharing his expertise in explainable AI methods with us.

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
