# Peer review of "Physics-aware Machine Learning for Glacier Ice Thickness Estimation: A Case Study for Svalbard"

_EGUsphere, 2024_

## Referee Comment (RC1)

**Review of "Physics-aware Machine Learning for Glacier Ice Thickness Estimation: A case Study for Svalbard" by Steidl and al.**

This paper proposes a framework based on Physics-aware machine learning to estimate the ice thickness distribution of grounded glaciers, applied to seven glaciers located in Svalbard. To achieve this, the authors incorporate mass conservation and other physical constraints into the cost function to be minimized during the training stage. They use ice thickness change and surface mass balance as primary constraints for the flux divergence. The authors conduct a glacier-wise cross-validation and discuss their results, and the cause of the rather-disappointing generalization in view to broader application.

This paper aligns with a series of recent works that explore the significant potential of machine learning techniques for ice thickness inference [e.g. Haq et al., 2021, Teisberg et al., 2021, Jouvet, 2023], including physics-informed deep learning [e.g. Riel and Minchew, 2023, 2022, Iwasaki and Lai, 2023, Jouvet and Cordonnier, 2023]. Among these, Physics-Informed Neural Networks (PINN) [Raissi et al., 2019] are particularly promising and represent an active research area.

Overall I find the approach interesting and I acknowledge the authors for their effort in presenting their results honestly. However, the work appears to be a work-in-progress, and lacks a clear picture of what should be done for making the method generalizable. This is partly due to a too a quickly-made and not very smooth writting (see e.g. my comments below about Section 2.3). The possible causes for the poor generalization results (LOGO CV) should be investigated (especially their relative importance) rather than listed, and I feel that this work could yield improvement in generalization already in this manuscript. I have a series of recommendations (encompassing the writing the paper and the methodology) that I hope will help the authors to improve the manuscript, which has the potential for a larger impact.

- In general, I find the description of the PINN rather inefficient. I focus here on Section 2.3, but my comments may be extrapolated to the entire paper. Do not expect TC readers to be familiar with all ML machinery, even less so with Physics-Informed ones. Therefore, you should provide a minimal background. Currently, section 2.3 is addressed solely to people with prior knowledge. At a minimum, briefly explain what a neural network is (a sequential composition of linear and nonlinear functions with optimizable weights). To my knowledge, this term can be intimidating, while it is not that complicated provided a minimal explanation. The current Section 2.3 mixes crucial information (I/O of the PINN) with more technical details (e..g activation functions, which are important but not essential for most readers unfamiliar with ML). I suggest distinguishing these two levels when rewriting this part to smooth the reading and target a broader audience. Consider moving ML technicalities to an appendix, and leave te ideas in the body of the paper. Another example: you mention "Fourier layers" but do not provide any rationale ( I would like to know the benefit of this). There are several ML-specific concepts (e.g., unlabeled data) that are not explained throughout the manuscript, which is a problem to maximize the audience of the paper to a general glaciological audience.

- I am not sure I understand: Do you feed your neural network with raster data grids (as suggested in Fig 1) or with large vectors of data at each coordinate along with the coordinate data? My question is whether you exploit the spatial structure of the data (I assume you have data on a raster structure grid). If not, I understand why you use a fully connneced

network; if you do, why not use a convolutional neural network designed to capture spatial relationships?

- The description of ice flow (Section 2.2) seems rather simplified. There are a couple of assumptions behind that are not clearly written down. Including a true high-order model here would be a great added value I think. You mention in line 274 that adding momentum conservation would be "technically easy," but I am less pessimistic than you about the claim that" it would complicate the optimization of the model." Instead, the functional associated with the Blatter-Pattyn model, for example, behaves relatively well with good convexity properties [Jouvet, 2016, Jouvet and Cordonnier, 2023], and could act as a physically-consistent, welcome smoother.

- In connection with my previous point, have you considered moving the surface velocity from the input of your PINN to the data? This would make sense if you are including momentum conservation. In the present case, can this be an option too? What is the motivation to insert the "observational" data in input of the PINN or as data constrained in the loss?

- The comparison (Section 4.2) to the two other products [Millan et al., 2022, Farinotti et al., 2019] is not a strong point. It tells us that the PINN lies within the range, which is not surprising as the two products differ significantly. This section could be moved to an appendix.

- Maybe consider applying your method first to a synthetic case where you can create a manufactured bedrock and dataset. Then, use your method to infer the ice thickness and compare it to the ground truth. This approach would help avoid issues related to data suspicion. In general, there are many possible causes for the lack of generalization, but there are strategies to isolate these causes that you could further explore through synthetic experiments.

- Section 5 provides a list of possible causes for the lack of generalization. However, it is hard to draw any conclusions. Some causes are more important than others. It would be helpful if you could prioritize these causes (and improvement items) by order of importance, from the most significant (with the largest potential for improvement) to the least significant. I feel that "Physical constraints" should be at the top of the list.

- Lines 264-266: You place a lot of trust in your Mass balance reconstruction, especially if it is not calibrated (line 264). Considering that this is a major constraint, I think this might be a significant cause of underperformance. Also, using a model for estimating the SMB (even a perfected one) is problematic, as your "observations" are not observations but modelled reconstructions. Have you considered using in-situ sparse measurements instead?

I have some additional specific comments:

- In the introduction, it would be good to ellaborate the existing literature on using ML for ice thickness inversion modeling [e.g. Haq et al., 2021, Teisberg et al., 2021, Jouvet, 2023] (line 21), as well as physics-informed deep learning applied to similar problems, such as inferring basal conditions (bedrock location or slipperiness) [e.g. Riel and Minchew, 2023, 2022, Iwasaki and Lai, 2023, Jouvet and Cordonnier, 2023] (lines 32-34). As this is a fast-evolving field, it would be good to check the latest papers, and possibly to complete.

- l 12: Not sure Millan et al. [2022] is the most appropriate reference for that.

- l 13: "Physics-based approaches ..." This sounds to be a very personal definition, consider a more appropriate one.

- l 22: "One avantage of data-driven approaches is a significant speed-up compared to physics-based models" : The computation speed-up has nothing to do with whether it is data-driven or physics-driven; it is the result of the efficiency of evaluating a neural network (especially on GPUs), irrepective of the training strategy: based on data [Jouvet et al., 2022] or on physics [Jouvet and Cordonnier, 2023]. Please correct.

- l 30-31: These two sentences are unclear to me : i) what means "data-efficient" in the context? ii) "boundary condition to solve the PDE", I think I understand what you mean (this would be a Dirichlet BC as you can enforce the solution to be close to a certain given value somewhere), but I'm not sure this is clear for all.

- l 255: "the loss landscape is highly complex", this is an unsual way to describe the lack of convexity the loss, which is not improved – I agree – by adding the number of constraints within the loss. I am not sure I found what optimizer you used (ADAM, SGD, RMSPROP, ?).

- Appendix B: I feel I have seen this exercise numerous times in textbooks, deriving a 0.8 ratio between vertically-averaged and surface velocity in the non-sliding SIA parallel slab case. I suggest you replace it a reference and use the space in the paper to better explain the ML part.

**References**

D. Farinotti, M. Huss, J. J. Fürst, J. Landmann, H. Machguth, F. Maussion, and A. Pandit. A consensus estimate for the ice thickness distribution of all glaciers on earth. *Nature Geoscience*, 12(3):168–173, 2019.

M. A. Haq, M. F. Azam, and C. Vincent. Efficiency of artificial neural networks for glacier ice-thickness estimation: A case study in western himalaya, india. *Journal of Glaciology*, 67(264): 671–684, 2021.

Y. Iwasaki and C.-Y. Lai. One-dimensional ice shelf hardness inversion: Clustering behavior and collocation resampling in physics-informed neural networks. *Journal of Computational Physics*, 492:112435, 2023.

G. Jouvet. Mechanical error estimators for shallow ice flow models. *Journal of Fluid Mechanics*, 807:40–61, 2016.

G. Jouvet. Inversion of a stokes glacier flow model emulated by deep learning. *Journal of Glaciology*, 69(273):13–26, 2023.

G. Jouvet and G. Cordonnier. Ice-flow model emulator based on physics-informed deep learning. *Journal of Glaciology*, pages 1–15, 2023.

G. Jouvet, G. Cordonnier, B. Kim, M. Lüthi, A. Vieli, and A. Aschwanden. Deep learning speeds up ice flow modelling by several orders of magnitude. *Journal of Glaciology*, 68(270):651–664, 2022.

R. Millan, J. Mouginot, A. Rabatel, and M. Morlighem. Ice velocity and thickness of the world's glaciers. *Nature Geoscience*, 15(2):124–129, 2022.

M. Raissi, P. Perdikaris, and G. E. Karniadakis. Physics-informed neural networks: A deep learning framework for solving forward and inverse problems involving nonlinear partial differential equations. *Journal of Computational physics*, 378:686–707, 2019.

B. Riel and B. Minchew. High-dimensional flow law parameter calibration and uncertainty quantification over antarctic ice shelves: a variational bayesian approach using deep learning. *Authorea Preprints*, 2022.

B. Riel and B. Minchew. Variational inference of ice shelf rheology with physics-informed machine learning. *Journal of Glaciology*, 69(277):1167–1186, 2023.

T. O. Teisberg, D. M. Schroeder, and E. J. MacKie. A machine learning approach to mass-conserving ice thickness interpolation. In *2021 IEEE International Geoscience and Remote Sensing Symposium IGARSS*, pages 8664–8667. IEEE, 2021.

---

## Referee Comment (RC2)

This work applies a Physics-Informed Neural Network (PINN) as a data-driven tool to estimate ice thickness across glaciers located on Spitsbergen in Svalbard. A physics-based loss function used in training the PINN is designed to penalize solutions diverging from a modified form of mass conservation. Additional physics-inspired loss functions are added relating components of the surface velocity arising from deformation of the ice and sliding at the base. The neural network is also provided with a number of other data inputs, including surface velocity, surface slope, elevation, positional parameters, and values providing assumed relationships between surface and depth-averaged velocities. The authors explore their results using a cross-validation scheme designed to avoid problems with spatial correlation.

PINNs have seen an increasing number of uses within glaciology in the past few years. Their intrinsic ability to mix together known physics with poorly calibrated constants and sparse and/or noisy measurements make them an appealing modeling tool for underdetermined problems. This work is novel in training a single PINN over an extremely large domain (effectively all of Spitsbergen) and in mixing a large number of physical constraints, physically-inspired constraints, and plausibly related data sources.

Two related challenges complicate the evaluation of results in this work. First, as with almost all ice-covered regions, direct measurements of ice thickness are sparse. Second, the dynamics of the glaciers are complex and poorly understood. In Svalbard, a number of complicating factors are at play:
1. Many glaciers are topographically constrained, making lateral drag important and complicating simplified models of ice dynamics.
2. Many glaciers are thought to be polythermal, often with a significant layer of temperate ice at the base overlain by cold ice (Sevestre et al., 2015)
3. Cold surface temperatures allow for the accumulation of thick firn layers with poorly constrained density (Pälli et al., 2017)

These complications are not unique to Svalbard, of course, but aspects of Svalbard's topography and geographic location make them especially notable here.

The authors frame the cross validation results in a way that seems somewhat disappointing. I am perhaps more optimistic than the authors about the results. In particular, I think the evaluation of a physically-based model on a glacier where no ice thickness data was provided is an unfair assessment of the model. The PINN proposed in this work is something of a hybrid between a data-driven estimator and a PDE solver. These two types of tools would be accessed in different ways. Additional consideration of appropriate evaluation mechanisms is probably needed.

Architecturally, I think this work is very interesting. There is a novel fusion of physics-based, physics-inspired, and non-physical relationships at work here. Unfortunately, the lack of explainability and the lack of a good ground-truth data source make it difficult to see a path to the results presented here significantly updating our thinking about Svalbard's glaciers.

Given this combination, I would encourage the authors to consider leaning into exploring the design of the PINN by, for example, exploring the importance of the various input fields or designing an experiment to consider the use of different ice physics approximations within this framework.

Specific comments below:

**PINN**

- It is not clear to me what the coordinate system is used to feed the network. Is it a standard projection? Are the coordinates consistent across all of Spitsbergen or are glaciers each on their own local coordinate system in some way?
- Why the current set of inputs to the neural network? It is not obvious to me, for example, why the area of the glacier should be included. In general, it would be interesting to know how including each input impacts the results.

**Physical Model**

- The way that deformation velocity, sliding velocity, depth-averaged velocity, and surface velocity are explained is somewhat confusing to me. My interpretation is that the authors are using a simplified physical model (Appendix B) to set a relationship between surface and depth averaged velocity, which you then qualitatively decide to loosen. Separately, they assume that sliding velocity and surface velocity are related by a pre-determined field. The network predicts deformation velocity only and evaluation of the mass conservation loss term is done by adding in sliding velocity according to the defined constant and the surface velocity.
    - What is the significance of the network outputting deformation velocity rather than directly depth-averaged velocity? Lines 61-62 seem to imply this is important, however it is not clear to me why. It seems to me that it is simply a choice between an extra calculation to compute mass balance and an extra calculation to compute the depth-averaged velocity bounds loss.
    - Lines 69-70 state that depth-averaged velocities are calculated for the x direction, y direction, and magnitude separately using different values of beta. Beta relates surface velocity to sliding velocity. In the simplified model of Appendix B, the sliding velocity must be in the same direction as the surface velocity, but different values of beta for x and y implies that the sliding velocity is in a different direction.
- Apart from stating that ice is assumed to be incompressible (Line 50), I saw no mention of the effects of unknown density of snow and firn. To my understanding, glaciers in Svalbard may have significant firn layers (Pälli et al., 2017). This contributes to uncertainty in the radar measurements (as the dielectric permittivity is dependent on density) and impacts the implied mass flux. This source of uncertainty should at least be discussed.
- In my view, the simplified ice dynamics of Appendix B may be insufficient for glaciers in Svalbard. I believe that the model selected ignores stresses from drag against the

sidewalls, which seem significant for the topographically constrained glaciers on Svalbard. Additionally, assuming A to be constant with depth seems like a stretch. Many glaciers are suspected to be polythermal and this has been proposed as a mechanism for the surge behavior seen in Svalbard (Sevestre et al., 2015). While the authors have excluded currently surging glaciers, the presence of this phenomenon implies to me that depth-dependent temperature may be an important part of glacier dynamics in this region. At a minimum, further discussion of this point is needed.

**OGGM-Processed Inputs**

- Are any of the input fields that are processed with OGGM interpolated by OGGM in any way? If they are interpolated following a similar physical model to yours, does this introduce a circularity?
- I think it would be helpful to discuss how the surface mass balance input is derived. It sounds like a model-derived value? There are quite a few weather stations in Svalbard. Has the model been validated? How does it perform?

**Training and Evaluation**

- The authors point out that data is highly correlated in space and thus they have used a cross-validation scheme based on leaving out an entire glacier at a time. I think that's a good approach to a challenging issue.
- With the above said, however, I do wonder if this is an overly harsh method of evaluation. The effect is that, in looking at Table 2, we're looking at glaciers where no ice thickness data was available, greatly diminishing the value of the mass conservation approach. Another approach might be to leave in only the highest (elevation) 20% of the ice thickness data and explore how well the PINN can use mass conservation to extrapolate this downstream.
- On Line 201, the authors state that the results suggest the model is overfitting. While this would be the conventional interpretation for a neural network, I think this is an overly critical interpretation for a PINN. Evaluating a PINN with no training data for the data loss function is sort of like evaluating a PDE solver with no boundary conditions. The analogy does not fully hold as the authors have also introduced some other inputs which can perhaps be used to guess at the ice thickness, but, in general, I think the authors may be too critical of their own results here.
- Later (Line 177), the authors mention a random split between training and validation data. Given the aforementioned spatial correlation problem, how is this validation dataset used? Is it meaningfully independent of the training data?

**Interpretation and Applications**

- It would be good to discuss the importance of ice thickness on Svalbard. This might depend on what you think your model is good at. For example, an improved estimate of total ice volume would be impactful for sea level rise projects. Improved fine-scale

ice-free topography might have more relevance to projecting the evolution of specific glaciers that are relevant to local communities.
- I would like to see discussion of what components of the inputs and loss function are most important. Many applications of PINNs largely use them as tools for solving PDEs where constraints, regularizations, or boundary conditions do not easily fit in conventional solvers. This work goes beyond that, feeding in multiple layers of data that is not directly incorporated into a physics-based loss term. This, of course, raises the question of which parts are most informative. A careful set of experiments exploring this would be very interesting.

**Typos and minor corrections**
- Line 10-11 - Ice flux is determined by more than simply ice thickness and surface slope under real world conditions. This should be clarified to not suggest that those two variables alone are sufficient.
- Line 69 - bracket is the wrong way around
- Figure 4 - Are the color scales saturating? If so, it would be good to show the clipping in a different color so we can see where the error exceeds +/- 100 m.
- In Table 2, comparing the first glacier's performance in-sample versus LOGO, the RMSD more than doubles while the MAPD decreases. Is this correct?

I enjoyed reading this work and believe it to be a promising avenue. I hope that these comments can help improve this manuscript.

---

## Author Response (AR1)

Dear Ben,

Thank you for giving us the opportunity to improve our manuscript. This letter combines our responses to both reviewers as well as a list of the changes to the original manuscript with line numbers.

The reviewers' comments are in black font and our answers are in blue font.

We are confident that the revisions we have made address the reviewers' concerns and enhance the overall quality of the manuscript. We sincerely hope that the revised version meets your and the reviewer's expectations, and we look forward to your response.

Best wishes,

Viola Steidl and co-authors

**Point-by-point responses to comments from Reviewer 1**

- In general, I find the description of the PINN rather inefficient. I focus here on Section 2.3, but my comments may be extrapolated to the entire paper. Do not expect TC readers to be familiar with all ML machinery, even less so with Physics-Informed ones. Therefore, you should provide a minimal background. Currently, section 2.3 is addressed solely to people with prior knowledge. At a minimum, briefly explain what a neural network is (a sequential composition of linear and nonlinear functions with optimizable weights). To my knowledge, this term can be intimidating, while it is not that complicated provided a minimal explanation. The current Section 2.3 mixes crucial information (I/O of the PINN) with more technical details (e.g. activation functions, which are important but not essential for most readers unfamiliar with ML). I suggest distinguishing these two levels when rewriting this part to smooth the reading and target a broader audience. Consider moving ML technicalities to an appendix, and leave te ideas in the body of the paper. Another example: you mention "Fourier layers" but do not provide any rationale ( I would like to know the benefit of this). There are several ML-specific concepts (e.g., unlabeled data) that are not explained throughout the manuscript, which is a problem to maximize the audience of the paper to a general glaciological audience.

  Thank you for pointing out the necessity to clarify machine learning terms. We totally agree that we should make our manuscript understandable to anyone without machine learning background. Therefore, we made significant revisions to the section. We added a brief description of neural networks to the section and streamlined the explanation of the I/O vectors:

  *"A neural network, also sometimes called multi-layered perceptron, consists of layers of connected nodes, also called neurons, where the connections each have an associated weight. At each node, the weighted outputs from each node of the previous layer are passed through a non-linear activation function (Goodfellow et al., 2016). By minimizing a loss the weights of the network are updated to make accurate predictions."*

  *"In a PINN model the loss is given by the residual of the PDE we want to solve. In theory, PINNs only require input features that are needed to calculate the derivatives in the PDE*

*(Raissi et al., 2018). In our work, we also provide the neural network with auxiliary data, that is related to glacier ice thickness but is not needed to solve the PDE. Therefore, we can exploit information from observable data as we would do it with a non-physics-aware neural network."*

*"The inputs to the model are vectors for each grid cell in the study region. They contain the spatial coordinates and surface velocities in x- and y-directions, and three β values to correct for basal sliding in x- and y-direction and in the magnitude. Additionally, the vectors contain auxiliary data like elevation, slope, the grid cell's distance to the border of its glacier, and the area of the glacier it belongs to."*

To better explain the Fourier embedding of the spatial coordinates we changed the name from "Fourier layer" to "Fourier feature encoding layer" and also added a description of the Fourier embedding:

*"The embedding of spatial coordinates was originally developed to overcome spectral bias in neural networks and speed up convergence in the reconstruction of images. It enables the network to learn high-frequency functions in low-dimensional problem domains."*

The rationale behind using the Fourier feature embedding is to speed up the convergence of the mass conserving loss that only relies on the derivatives w.r.t. the spatial coordinates. Figure 1 (not included in the manuscript) shows that the Fourier feature embedding clearly makes the mass conservation loss drop faster, whereas it would not be improved at all without the Fourier feature embedding.

The concept of labelled and unlabelled data is now also explained:

*"We refer to the points with ice thickness measurements as labelled, whereas points without being referred to as unlabelled."*

[Figure]

*Figure 1* Comparison of mass conservation loss with and without Fourier feature embedding layer

- I am not sure I understand: Do you feed your neural network with raster data grids (as suggested in Fig 1) or with large vectors of data at each coordinate along with the coordinate data? My question is whether you exploit the spatial structure of the data (I assume you have

data on a raster structure grid). If not, I understand why you use a fully connected network; if you do, why not use a convolutional neural network designed to capture spatial relationships?

Thank you for bringing up that this could be misunderstood. As mentioned before we now clarified that the training data are vectors of data at each point of the grid.

*"The inputs to the model are vectors for each grid cell in the study region."*

The spatial structure is not exploited with a convolutional network yet, but we agree that this is an interesting follow-up.

- The description of ice flow (Section 2.2) seems rather simplified. There are a couple of assumptions behind that are not clearly written down. Including a true high-order model here would be a great added value I think. You mention in line 274 that adding momentum conservation would be "technically easy," but I am less pessimistic than you about the claim that "it would complicate the optimization of the model." Instead, the functional associated with the Blatter-Pattyn model, for example, behaves relatively well with good convexity properties [Jouvet, 2016, Jouvet and Cordonnier, 2023], and could act as a physically-consistent, welcome smoother.

Thank you for your comment. As this also came up in the second Review letter we revised Section 2.2 and included the assumptions to the SIA:

*"There are models with different degrees of approximations to the full Navier-Stokes equations to describe ice flow. The simplest one, the shallow ice approximation (SIA) assumes lamellar flow, so the driving forces are entirely opposed by basal drag. It neglects lateral shear and longitudinal stresses and the rate factor A from Glen's flow law is taken to be constant with depth* (van der Veen, 2013)*."*

We agree that including a higher-order model could provide better estimates of the velocity profile with depth. However, to apply these models we would need to make further assumptions, for example, about the ice viscosity and how it varies with depth or the amount of basal drag/drag from the sidewalls of the glaciers. Indeed, Rückamp et al. (2022) identify this as an issue with the Blatter-Pattyn approximation to full Stokes solutions. We want to emphasise here, that our study is a proof of concept rather than a definitive analysis. We identify several areas for improvement in future work and a higher order model for surface to depth average velocity is one possibility but, likely, not the first order issue for improving the results, which we believe are more sensitive to i) the quality of the input data, ii) the SMB estimates used and iii) estimation of basal velocities. We discuss how each of these issues could be addressed in future work.

Also we clarified our claim about adding momentum conservation being technically easy. We meant to say that adding another component in the loss function is technically easy to do, as it is just adding another term. However, supporting the correct evaluation of the loss requires detailed knowledge about parameters like the viscosity of ice. We now rewrote the sentence to make it less ambiguous:

*"While this is technically easy to do, it comes at the cost of introducing uncertainties from approximating required parameters. We would have to make assumptions about ice viscosity and resistance from the bedrock, for example."*

Thanks again for bringing up that the way we phrased it could be misunderstood.

- In connection with my previous point, have you considered moving the surface velocity from the input of your PINN to the data? This would make sense if you are including momentum conservation. In the present case, can this be an option too? What is the motivation to insert the "observational" data in input of the PINN or as data constrained in the loss?

  We assume with 'moving the surface vel from input of your PINN to the data?' you suggest having the surface velocity in the target vector instead of the Input vector. In fact, this would be an option, too and Teisberg et al. set up their model exactly in this way. However, as the (surface) velocity is actually an important predictor of the ice thickness, we decided to leave it in the input vector.

  The idea behind having the apparent mass balance only in the target vector is that we are not confident about the quality of the mass balance data as it is modelled from a simple model. Therefore, we did not want to have it as an input that would give the mass balance data more weight as compared to only introducing it with the soft constraint of the mass conservation loss.

- The comparison (Section 4.2) to the two other products [Millan et al., 2022, Farinotti et al., 2019] is not a strong point. It tells us that the PINN lies within the range, which is not surprising as the two products differ significantly. This section could be moved to an appendix.

  We agree it is not a strong point to prove the correctness of the PINN's ice thickness estimate. However, we think it is informative to show how the estimate compares to other ice thickness estimates. Therefore, we would like to keep it in the Results section.

- Maybe consider applying your method first to a synthetic case where you can create a manufactured bedrock and dataset. Then, use your method to infer the ice thickness and compare it to the ground truth. This approach would help avoid issues related to data suspicion. In general, there are many possible causes for the lack of generalization, but there are strategies to isolate these causes that you could further explore through synthetic experiments.

  We totally agree that applying the method to a perfect synthetic case would be the optimal setting to test the method and research causes for bad generalization. This would be an interesting follow-up exercise but is, by no means, a trivial exercise for the following reasons. The design of the experiment and the design of the synthetic data are crucial in our view. For example, do we use a Full Stokes model, Blatter-Pattyn or some other approximation. Which kinds of glaciers should be modelled with what kind of glacier bed? How to best sample a variety of glaciers? We would need a range of SMB profiles and, presumably, a range of bedrock thermal regimes from fully frozen, partially frozen to temperate and so on. A synthetic data approach would certainly allow us to explore how uncertainties and assumptions influence the robustness of the solution but would be a substantial effort in its own right.

Nevertheless, we agree that applying the approach to a synthetic dataset would be ideal to better evaluate the PINN model and its strengths or weaknesses.

- Section 5 provides a list of possible causes for the lack of generalization. However, it is hard to draw any conclusions. Some causes are more important than others. It would be helpful if you could prioritize these causes (and improvement items) by order of importance, from the most significant (with the largest potential for improvement) to the least significant. I feel that "Physical constraints" should be at the top of the list.

  We agree that listing the potential causes for bad generalization is not ideal. However, it is certainly not trivial to prioritize the possible causes. We would, for example, argue that input data quality plays a huge, perhaps dominant, role. The relative weighting of data loss and the physics-aware losses, also in close relation to the amount of noise in the measurement data, has a significant impact on the convergence of the PINN (Iwasaki and Lai, 2023). Since in our model the quality/label uncertainty is not yet taken into account, we believe that this could be one way to improve the model. However, improved SMB and basal velocity estimation will also be important, as we state. For the latter, there are several approaches that could be adopted such as using winter-only velocities or by examining the seasonal cycle in velocities.

  We agree that physical constraints play a significant role but the significance will likely vary by glacier. To address this concern we have indicated, qualitatively, the factors that would significantly improve the solution.

- Lines 264-266: You place a lot of trust in your Mass balance reconstruction, especially if it is not calibrated (line 264). Considering that this is a major constraint, I think this might be a significant cause of underperformance. Also, using a model for estimating the SMB (even a perfected one) is problematic, as your "observations" are not observations but modelled reconstructions. Have you considered using in-situ sparse measurements instead?

  Yes, we thought about using observations but as the objective is to evaluate the mass conservation at each point of the grid, we need to fall back to a mass balance product that is available for the entire study area. The mass balance reconstruction that we are using is actually calibrated on observational data (https://docs.oggm.org/en/stable/mass-balance-monthly.html).

  However, maybe in a follow-up work, it would be worthwhile to include another loss component where the residual to mass conservation is calculated from in situ SMB measurements wherever they are available, just like the data loss is evaluated only where ice thickness measurements are available. Thanks for making this suggestion.

I have some additional specific comments:

- In the introduction, it would be good to ellaborate the existing literature on using ML for ice thickness inversion modeling [e.g. Haq et al., 2021, Teisberg et al., 2021, Jouvet, 2023] (line 21), as well as physics-informed deep learning applied to similar problems, such as inferring basal conditions (bedrock location or slipperiness) [e.g. Riel and Minchew, 2023, 2022, Iwasaki and Lai, 2023, Jouvet and Cordonnier, 2023] (lines 32-34). As this is a fast-evolving field, it would be good to check the latest papers, and possibly to complete.

Thank you for providing further literature that should be included. We extended the literature review to make it more complete. We hope this meets your expectations.

- l 12: Not sure Millan et al. [2022] is the most appropriate reference for that.
  Thanks for pointing that out, we apologize for the mistake and changed the reference to Welty et al. 2020.

- l 13: "Physics-based approaches …"   This sounds to be a very personal definition, consider a more appropriate one.
  Agreed, we changed the sentence such that it cannot be misunderstood as a definition anymore: *"There are physics-based and process-based approaches that aim to reconstruct glacier ice thicknesses from in situ data and ice dynamical considerations."*

- l 22: "One avantage of data-driven approaches is a significant speed-up compared to physics based models": The computation speed-up has nothing to do with whether it is data-driven or physics-driven; it is the result of the efficiency of evaluating a neural network (especially on GPUs), irrepective of the training strategy: based on data [Jouvet et al., 2022] or on physics [Jouvet and Cordonnier, 2023]. Please correct.

  Thank you for the correction. We changed the sentence to clarify that we are talking about data-driven machine learning methods that are fast to optimize and evaluate: *"One advantage of machine learning approaches is their efficient optimization and evaluation compared to process-based models* (Jouvet et al., 2022)*."*

- l 30-31: These two sentences are unclear to me : i) what means "data-efficient" in the context? ii) "boundary condition to solve the PDE", I think I understand what you mean (this would be a Dirichlet BC as you can enforce the solution to be close to a certain given value somewhere), but I'm not sure this is clear for all.
  Thank you very much for bringing up that this is not clear. With 'data-efficient' we meant to describe that we are less dependent on ground truth data because we are also relying on physical constraints. We took this out to avoid misunderstanding. Also, as you rightfully mentioned the term boundary condition might be misleading as the data loss is not exactly a condition that we set on the boundary of the domain but rather an "internal constraint" that helps find a solution to the PDE. We also changed this wording in the manuscript: *"Additional ground truth data can be used to compute a data loss that acts as an internal condition to constraining solutions to the PDE."*

- l 255: "the loss landscape is highly complex", this is an unsual way to describe the lack of convexity the loss, which is not improved – I agree – by adding the number of constraints within the loss. I am not sure I found what optimizer you used (ADAM, SGD, RMSPROP, ?).
  We apologize for not including this information in the manuscript before. We used the Adam optimizer and added the information to the new Appendix Section on the architecture of the model.

- Appendix B: I feel I have seen this exercise numerous times in textbooks, deriving a 0.8 ratio between vertically-averaged and surface velocity in the non-sliding SIA parallel slab case. I suggest you replace it a reference and use the space in the paper to better explain the ML part.

We agree that this is often described in textbooks, but we would like to keep the derivation as an explanation of where our lower bound to the depth-averaged velocity estimation comes from and also which assumptions have been made.

Goodfellow, I., Bengio, Y., and Courville, A.: Deep Learning, MIT Press, 2016.

Iwasaki, Y. and Lai, C.-Y.: One-dimensional ice shelf hardness inversion: Clustering behavior and collocation resampling in physics-informed neural networks, J. Comput. Phys., 492, 112435, https://doi.org/10.1016/j.jcp.2023.112435, 2023.

Jouvet, G., Cordonnier, G., Kim, B., Lüthi, M., Vieli, A., and Aschwanden, A.: Deep learning speeds up ice flow modelling by several orders of magnitude, J. Glaciol., 68, 651–664, https://doi.org/10.1017/jog.2021.120, 2022.

Raissi, M., Perdikaris, P., and Karniadakis, G. E.: Physics-informed neural networks: A deep learning framework for solving forward and inverse problems involving nonlinear partial differential equations, J. Comput. Phys., 378, 686–707, https://doi.org/10.1016/j.jcp.2018.10.045, 2018.

Rückamp, M., Kleiner, T., and Humbert, A.: Comparison of ice dynamics using full-Stokes and Blatter–Pattyn approximation: application to the Northeast Greenland Ice Stream, The Cryosphere, 16, 1675–1696, https://doi.org/10.5194/tc-16-1675-2022, 2022.

van der Veen, C. J.: Fundamentals of Glacier Dynamics, Second edition., CRC Press, 2013.

**Point-by-point response to comments from Reviewer 2**

Two related challenges complicate the evaluation of results in this work. First, as with almost all ice-covered regions, direct measurements of ice thickness are sparse. Second, the dynamics of the glaciers are complex and poorly understood. In Svalbard, a number of complicating factors are at play:

1. Many glaciers are topographically constrained, making lateral drag important and complicating simplified models of ice dynamics.

2. Many glaciers are thought to be polythermal, often with a significant layer of temperate ice at the base overlain by cold ice (Sevestre et al., 2015)

3. Cold surface temperatures allow for the accumulation of thick firn layers with poorly constrained density (Pälli et al., 2017)

These complications are not unique to Svalbard, of course, but aspects of Svalbard's topography and geographic location make them especially notable here. The authors frame the cross validation results in a way that seems somewhat disappointing. I am perhaps more optimistic than the authors about the results. In particular, I think the evaluation of a physically-based model on a glacier where no ice thickness data was provided is an unfair assessment of the model. The PINN proposed in this work is something of a hybrid between a data-driven estimator and a PDE solver. These two types of tools would be accessed in different ways. Additional consideration of appropriate evaluation mechanisms is probably needed.

Architecturally, I think this work is very interesting. There is a novel fusion of physics-based, physics-inspired, and non-physical relationships at work here. Unfortunately, the lack of explainability and the lack of a good ground-truth data source make it difficult to see a path to the results presented here significantly updating our thinking about Svalbard's glaciers. Given this combination, I would encourage the authors to consider leaning into exploring the design of the PINN by, for example, exploring the importance of the various input fields or designing an experiment to consider the use of different ice physics approximations within this framework.

Thank you for your assessment of our work. We agree that there is a lot of potential that can be explored in follow-up studies and you make a number of useful suggestions

However, we want to reiterate that this is a proof-of-concept (PoC) study to assess the viability of a PINN approach for a long-standing challenge in glaciology, and we use Svalbard as a test case, partly for the reasons you mention above related to the range of flow conditions and glacier geometries and partly because it is one of few areas with relatively good coverage from observations and other estimates. Indeed, we compare our solution with three others and show that it is not so easy to infer that one of those four is "preferable". Our intention is not to shed new light on Svalbard ice thickness but on the potential of physics-informed ML for this problem. We have made this more explicit and clearer in the Introduction to avoid any ambiguity about our aims and focus:

*"As a proof of concept, we include all non-surging glaciers in Spitsbergen, Barentsøya, and Edgeøya in Svalbard to show that it is possible to use a PINN architecture for an entire region."*

We also repeat the statement in the Conclusion:

*"This serves as a proof of concept that physics-informed models can not only be applied to one single closed system but, together with auxiliary data, can make meaningful predictions for entire regions."*

Specific comments below:

**PINN**

- It is not clear to me what the coordinate system is used to feed the network. Is it a standard projection? Are the coordinates consistent across all of Spitsbergen or are glaciers each on their own local coordinate system in some way?
  We apologize that this was not clear in the manuscript. The coordinates are consistent across the entire study region (now mentioned in the manuscript: "*The coordinates of the individual grids are all transformed to the same projection.*"), we used the EPSG:25832 projection (which is mentioned in the model configuration file in the linked github repository).

- Why the current set of inputs to the neural network? It is not obvious to me, for example, why the area of the glacier should be included. In general, it would be interesting to know how including each input impacts the results.
  In general, the features are added because they have some relation to the glacier ice thickness and are available through OGGM. It is left to the neural network to find relations between the input and the target.
  The area of the glacier is added to the input vector as area/volume scaling is a well established approach that has been used in the past to estimate thickness (e.g. Bahr et al., 1997).
  We agree that a detailed analysis of feature importance would be interesting for explainability of the results. Therefore, we conducted a quantitative analysis of the feature

importance through SHAP (SHapley Additive exPlanations) using Captum's (Kokhlikyan et al., 2020) implementation of the DeepLIFT SHAP algorithm (Lundberg and Lee, 2017) and included this in the Appendix.

The framework explains feature contributions to the model prediction, usually for purely data-driven machine learning. It comes with some limitations: all features should ideally be independent of each other. This is clearly not the case for our features. Nevertheless, we used this framework as it is a standard approach often used in machine learning.

We conducted the SHAP analysis on all of the seven LOGO CV models. Figure 1 shows the mean absolute SHAP values: high SHAP values signify a high impact on the output of the model; low absolute values signify a low impact. For our PINN, the spatial coordinates are by far the most important. This is expected as they define the domain of our solution to the mass conservation PDE.

[Figure]

*Figure 2* Mean of the approximated SHAP values for all seven LOGO CV m*odels.*

Besides the spatial coordinates the slope has the biggest impact on the prediction. Figure 2 provides a more detailed view of the impact of features. For every point in the dataset it shows how it impacts the predicted thickness.

The colour signifies the feature value of the individual data points: red signifies a relatively high value (within the range of the feature values), and blue signifies a relatively low value. For example, datapoints with low values for slope are more likely to lead to high values for the predicted ice thickness, while high values are less likely to impact the thickness prediction or rather decrease it.

This is what we would expect given that ice thickness and slope are indirectly proportionally related in the SIA: flat slopes lead to thicker ice.

The SHAP values for the distance-to-border feature tell us that the model thinks that at the border the ice thickness should be smaller than within the glacier.

For the surface velocity values the interpretation is less clear, also because we only see the component-wise features. High surface velocities do not seem to have much impact on the ice thickness prediction, although, following glacier physics, they should have a strong influence on ice thickness.

Overall, the SHAP analysis is a good tool for deeper insights into the correlations between feature values and model prediction. However, we should be careful not to overestimate the explanatory power, especially when dealing with correlated features.

The analysis depends very much on the dataset (we chose the validation datasets to conduct the analysis). Therefore, the results can only show the impact of the features on the ice thickness prediction for our specific dataset and model setup. We can not derive universal feature importance from the analysis, let alone find causal relationships.

[Figure]

*Figure 3* SHAP values for each datapoint by feature. The colour shows the relative value the feature takes for each datapoint.

**Physical Model**

- The way that deformation velocity, sliding velocity, depth-averaged velocity, and surface velocity are explained is somewhat confusing to me. My interpretation is that the authors are using a simplified physical model (Appendix B) to set a relationship between surface and depth averaged velocity, which you then qualitatively decide to loosen. Separately, they assume that sliding velocity and surface velocity are related by a pre-determined field. The network predicts deformation velocity only and evaluation of the mass conservation loss term is done by adding in sliding velocity according to the defined constant and the surface velocity.

  Thank you very much for your comment on this section. Reviewing the section we found that there was some confusion in it. Therefore, *we rewrote the whole Section 2.2*. The amount of basal sliding is introduced in the equation with the factor β that is estimated from the ratio of surface slope and surface velocity, following Millan et al., (2022).

  If there is no basal sliding at all we estimate the depth-averaged velocity to be smaller than the surface velocity. We set a lower boundary assuming that depth-averaged velocity won't be any smaller than 70% of the surface velocity.

  If the entire surface velocity would be due to basal sliding then the depth-averaged velocity is equal to the surface velocity. The estimate should reflect that the more basal sliding we have the closer the depth-averaged velocity will be to the surface velocity.

  - What is the significance of the network outputting deformation velocity rather than directly depth-averaged velocity? Lines 61-62 seem to imply this is important, however it is not clear to me why. It seems to me that it is simply a choice between an extra calculation to compute mass balance and an extra calculation to compute the depth-averaged velocity bounds loss.

    We apologize for the confusion in this subsection. In the revised model we directly estimate depth-averaged velocity and updated the description of the calculation of the depth-averaged loss:

$$\mathcal{L}_{vel} = \begin{cases} (v_s - \bar{v})^2 & \text{if } \bar{v} > v_s \\ (v_s(l_{lower} + (1 - l_{lower})\beta) - \bar{v})^2 & \text{if } \bar{v} < v_s(l_{lower} + (1 - l_{lower})\beta) \\ 0 & \text{else} \end{cases} \quad \begin{aligned} &\text{with } \bar{v} \in \{\bar{v}_x, \bar{v}_y, \bar{v}_{mag}\} \\ &\text{and } \beta \in \{\beta_x, \beta_y, \beta_{mag}\} \end{aligned}$$

  - Lines 69-70 state that depth-averaged velocities are calculated for the x direction, y direction, and magnitude separately using different values of beta. Beta relates surface velocity to sliding velocity. In the simplified model of Appendix B, the sliding velocity must be in the same direction as the surface velocity, but different values of beta for x and y implies that the sliding velocity is in a different direction.

    We have 3 different β values, that are used separately in the component-wise/magnitude estimate of depth-averaged velocity. The β values are derived from the velocity in x- and y-direction and the magnitude separately, so it is ensured that the estimate of the amount of sliding is along the same direction as the surface velocity. To clarify this we added it to the description in Section 3.2:

    *"For each point, we compute three β values from the surface velocities in the x- and y-direction and the magnitude of the surface velocity."*

- Apart from stating that ice is assumed to be incompressible (Line 50), I saw no mention of the effects of unknown density of snow and firn. To my understanding, glaciers in Svalbard may have significant firn layers (Pälli et al., 2017). This contributes to uncertainty in the radar measurements (as the dielectric permittivity is dependent on density) and impacts the implied mass flux. This source of uncertainty should at least be discussed.

  *Thank you for mentioning that, we added it as a source of uncertainty in the discussion part.*
  *"Ice thickness measurements from ice-penetrating radar, for example, are subject to errors due to varying density of glacier ice but also due to unknown thickness of snow and firn layers (Lindbäck et al., 2018)."*
  *In general, we think that the uncertainties of the measurement data play a big role in the model performance and we emphasize that throughout the manuscript. Some of the in situ data points are also with a given uncertainty that in a follow-up would be interesting to include.*

- In my view, the simplified ice dynamics of Appendix B may be insufficient for glaciers in Svalbard. I believe that the model selected ignores stresses from drag against the sidewalls, which seem significant for the topographically constrained glaciers on Svalbard. Additionally, assuming A to be constant with depth seems like a stretch. Many glaciers are suspected to be polythermal and this has been proposed as a mechanism for the surge behavior seen in Svalbard (Sevestre et al., 2015). While the authors have excluded currently surging glaciers, the presence of this phenomenon implies to me that depth-dependent temperature may be an important part of glacier dynamics in this region. At a minimum, further discussion of this point is needed.

  *We agree that our model neglects a lot of physical processes and uses very simplified physical descriptions. You are right that our model does not specifically account for drag against the sidewalls. The estimate of the ice flow and velocities is not very restricted in the sense that we only set an upper and a lower limit for the depth-averaged velocity. Within those boundaries, the model can freely estimate the depth-averaged velocity.*
  *This is a design choice that we explicitly made to avoid choosing a physical model with parameters like the viscosity of ice that we cannot be certain about and would introduce new uncertainties.*
  *We added this to the Discussion section to better emphasize this point. Also, we included your point that A is actually temperature-dependent in the Discussion and not only in the derivation of the lower bound for the velocity estimate.*
  *"This approximation does not account for lateral drag and assumes the creep coefficient A to be temperature-independent. However, many glaciers in Svalbard are believed to be polythermal (Glasser, 2011). Therefore, the estimate of the depth-averaged velocity might have another source of uncertainty that is challenging to quantify."*

**OGGM-Processed Inputs**

- Are any of the input fields that are processed with OGGM interpolated by OGGM in any way? If they are interpolated following a similar physical model to yours, does this introduce a circularity?

  *Yes, the input collected through OGGM are interpolated to the grids of the glaciers. To our best knowledge, the source code of OGGM uses transformations that reproject and scale the data to the glacier grids but do not use a physical model. Mostly the data is reprojected using methods from rasterio or salem libraries. (Example for reprojection of the dh/dt data from Hugonnet et al., 2021: https://github.com/OGGM/oggm/blob/master/oggm/shop/hugonnet_maps.py#L12).*
  *We clarified that in the manuscript:*

*"OGGM reprojects and scales the data for each glacier to the glacier grids. We collect these data and transform the coordinates from the individual grids into a common projection."*

- I think it would be helpful to discuss how the surface mass balance input is derived. It sounds like a model-derived value? There are quite a few weather stations in Svalbard. Has the model been validated? How does it perform?
  You are right; the surface mass balance is modelled with OGGM's ConstantMassBalance model. The model is calibrated from geodetic mass balance measurements and computes mass balance according to the elevation of the data point (https://docs.oggm.org/en/stable/mass-balance-monthly.html). We have not validated the model on in situ data but already the OGGM documentation states that "more physical approaches are possible". Therefore, in the Discussion we mention that improving the estimate of the apparent mass balance calculation is one of the more important tasks to improve the performance of our PINN model.

**Training and Evaluation**

- The authors point out that data is highly correlated in space and thus they have used a cross-validation scheme based on leaving out an entire glacier at a time. I think that's a good approach to a challenging issue.
  Thank you.

- With the above said, however, I do wonder if this is an overly harsh method of evaluation. The effect is that, in looking at Table 2, we're looking at glaciers where no ice thickness data was available, greatly diminishing the value of the mass conservation approach. Another approach might be to leave in only the highest (elevation) 20% of the ice thickness data and explore how well the PINN can use mass conservation to extrapolate this downstream.
  We agree on the fact that it might be a harsh method to evaluate the model, but given that we are also predicting ice thicknesses for glaciers where no in situ thickness measurements are available it is crucial to know what the expected accuracy is on those glaciers. The LOGO cross validation, therefore, tells us that we should not be too certain about the predictions the PINN makes on any glacier without any given in situ measurements.
  Training on only ice thickness measurements above a certain altitude is an interesting approach to evaluating performance additionally. For generalizability, we think the LOGO CV is the most important.

- On Line 201, the authors state that the results suggest the model is overfitting. While this would be the conventional interpretation for a neural network, I think this is an overly critical interpretation for a PINN. Evaluating a PINN with no training data for the data loss function is sort of like evaluating a PDE solver with no boundary conditions. The analogy does not fully hold as the authors have also introduced some other inputs which can perhaps be used to guess at the ice thickness, but, in general, I think the authors may be too critical of their own results here.
  We agree that the task is challenging if there are no boundary conditions or measurements to constrain the model predictions more. However, we want to be clear that the model cannot yet predict ice thickness for glaciers without in situ measurements with the same accuracy as for glaciers where we provide ice thickness measurements.
  As you correctly mentioned, the model actually has other input features that it can use to learn the distribution of ice thicknesses. To us this seems very much like an overfitting problem where the model fits very well to the training data and does not generalize well to regions where the labels have not been in the training data.

In theory the physics-aware loss components should take over in these areas. Therefore, improving this overfitting problem is a bit trickier than in purely-data driven machine learning models. On top of all the machine learning reasons that could lead to overfitting, there are also a couple of issues related to the physics-aware part of the model, like finding the optimal balance between loss components to enforce physical consistency that the noisy training data probably cannot even provide. Apart from that, we think 'overfitting' describes the situation quite well.

- Later (Line 177), the authors mention a random split between training and validation data. Given the aforementioned spatial correlation problem, how is this validation dataset used? Is it meaningfully independent of the training data?
  The random split is not meaningfully independent of the training data because of their spatial closeness to the training data. We clarified this in the manuscript by adding
  *"The training and validation data are spatially correlated. Therefore, the in-sample evaluation of the model probably overestimates its performance."*
  However, the split is useful to compare the results from the LOGO CV against each other and to the performance on the test glaciers. Since the in-sample performances do not significantly differ from model to model, we can be sure that the method is at least robust to leaving out thickness data of entire glaciers.

**Interpretation and Applications**

- It would be good to discuss the importance of ice thickness on Svalbard. This might depend on what you think your model is good at. For example, an improved estimate of total ice volume would be impactful for sea level rise projects. Improved fine-scale ice-free topography might have more relevance to projecting the evolution of specific glaciers that are relevant to local communities.
  We think that, since our work is more on the methodological side of estimating ice thickness, discussing the importance of ice thickness on Svalbard's glaciers would shift the focus too much away from the core of the work. Please see our response to your first comment above. In addition, our paper has been submitted to TC where we believe readers will understand why ice thickness is important. Nonetheless, we have added the following sentences to the intro which we believe is a strong justification:
  *"Ice thickness is the single most important input for modelling the dynamics of an ice mass because surface velocity is proportional to the fourth power of thickness. Combined with surface elevation, it provides bed topography, also key for modelling flow."*

  Once the method produces robust and reliable results for unseen glaciers, it could help both in improving the estimate of total ice volume and also the fine-scale ice-free topography (as we can choose the grid resolution as fine as we need them).

- I would like to see discussion of what components of the inputs and loss function are most important. Many applications of PINNs largely use them as tools for solving PDEs where constraints, regularizations, or boundary conditions do not easily fit in conventional solvers. This work goes beyond that, feeding in multiple layers of data that is not directly incorporated into a physics-based loss term. This, of course, raises the question of which parts are most informative. A careful set of experiments exploring this would be very interesting.

  We agree that the importance of the individual loss components is also interesting to quantify, just as the importance of the loss components (already discussed earlier). However, it is tricky to

evaluate, as we are dealing with unevenly distributed, correlated, noisy in situ measurements as labels to evaluate the PINN performance.

Despite this, we ran experiments in which we set the weight of each of the loss components to zero one after another. We then compared the scores to the scores of the reported model. To do so we calculated a relative RMSD as relative RMSD = $RMSD_{reported}$-RMSD/$RMSD_{reported}$. The relative RMSD will be positive if the score improves, and negative if the score gets worse by setting the weight of a loss component to 0.

The relative differences vary among the scores for the test glaciers but are below 5% on average, as shown in Table 1. However, it is interesting to see that, while the scores on the **in-sample** validation data improve on average when switching off the physics-aware loss components (see Table 2), the scores on the **out-of-sample** test glaciers get worse on average.

This matches our intuition that the model is overfitting on the in situ ice thickness data that we provide it with during training. The physics-aware loss components act like a regularization while demanding physical consistency. From this experiment, it looks like the loss to bound the estimate of the depth-averaged velocity is the most important component.

This intuitively makes sense as a wrong estimate of depth-averaged velocity directly influences the mass conservation loss as well. With corrupted depth-averaged velocities, the mass conservation loss will not be able to enforce physical consistency.

Nevertheless, we want to emphasize again that the configuration of the loss weights is certainly not optimal, as we discussed in Section 5.3, so there might be another distribution of importance if all the loss components are better balanced. Also, as already mentioned, this experiment depends a lot on the dataset, so the importance of loss components also only applies to this specific study.

We added this Discussion to the Appendix.

For the Discussion of the importance of input features, please refer to our answer to your earlier question in the Section "PINN".

**Relative RMSD for the scores from the out-of-sample test glaciers of the seven models (no in situ ice thicknesses for those glaciers were in the training dataset)**

| Relative RMSD For test glaciers | No MC | Vel_loss | Smoothness | Negative thickness |
|---|---|---|---|---|
| RGI60-07.00240 | -0,091 | -0,195 | -0,013 | -0,013 |
| RGI60-07.00344 | -0,018 | -0,036 | 0,000 | 0,018 |
| RGI60-07.00496 | -0,132 | -0,184 | 0,105 | -0,053 |
| RGI60-07.00497 | 0,000 | -0,087 | 0,000 | -0,022 |
| RGI60-07.01100 | 0,025 | -0,125 | -0,150 | 0,000 |
| RGI60-07.01481 | 0,000 | 0,072 | -0,157 | -0,024 |
| RGI60-07.01482 | 0,008 | 0,032 | -0,048 | -0,048 |
| Relative RMSD Mean | **-0,030** | **-0,075** | **-0,038** | **-0,020** |

*Table 1* Test glacier relative RMSD

**Relative RMSD for the scores from the in-sample validation data for the seven models**

| Relative RMSD In-sample Val | No MC | Vel_loss | Smoothness | Negative thickness |
|---|---|---|---|---|
| RGI60-07.00240 | -0,033 | 0,000 | -0,033 | -0,033 |

| | | | | |
|---|---|---|---|---|
| **RGI60-07.00344** | 0,032 | 0,032 | 0,065 | 0,000 |
| **RGI60-07.00496** | 0,000 | 0,030 | 0,061 | 0,000 |
| **RGI60-07.00497** | 0,033 | 0,000 | 0,033 | 0,000 |
| **RGI60-07.01100** | 0,032 | 0,032 | 0,032 | 0,000 |
| **RGI60-07.01481** | 0,000 | 0,033 | 0,033 | -0,033 |
| **RGI60-07.01482** | 0,034 | 0,034 | 0,034 | 0,000 |
| **Relative RMSD Mean** | **0,014** | **0,023** | **0,032** | **-0,010** |

*Table 2* In-sample validation relative RMSD

**Typos and minor corrections**

- Line 10-11 - Ice flux is determined by more than simply ice thickness and surface slope under real world conditions. This should be clarified to not suggest that those two variables alone are sufficient.
  Thanks for the remark, we meant to emphasize that ice thickness is most important to reliably **model** ice dynamics. Therefore, we changed the sentence to
  *"Glacier ice thickness is a fundamental variable required for modelling the evolution of a glacier."*
- Line 69 - bracket is the wrong way around
  The bracket opening to the left should signal that 0 is outside the interval as it is not a possible value for the parameter. The section was rewritten without the bracket now.
- Figure 4 - Are the color scales saturating? If so, it would be good to show the clipping in a different color so we can see where the error exceeds +/- 100 m.
  Thanks for the remark; we updated the figure:

[Figure]

- In Table 2, comparing the first glacier's performance in-sample versus LOGO, the RMSD more than doubles while the MAPD decreases. Is this correct?
  Yes, this is correct. This is because the glacier is one of the thicker glaciers. Therefore, a high RMSD might not directly lead to a high MAPD as the MAPD is the error relative to the value of the true ice thickness.

I enjoyed reading this work and believe it to be a promising avenue. I hope that these comments can help improve this manuscript.

Thank you again for all your comments, we think it greatly helped to improve the manuscript.

Bahr, D. B., Meier, M. F., and Peckham, S. D.: The physical basis of glacier volume-area scaling, J. Geophys. Res. Solid Earth, 102, 20355–20362, https://doi.org/10.1029/97JB01696, 1997.

Hugonnet, R., McNabb, R., Berthier, E., Menounos, B., Nuth, C., Girod, L., Farinotti, D., Huss, M., Dussaillant, I., Brun, F., and Kääb, A.: Accelerated global glacier mass loss in the early twenty-first century, Nature, 592, 726–731, https://doi.org/10.1038/s41586-021-03436-z, 2021.

Kokhlikyan, N., Miglani, V., Martin, M., Wang, E., Alsallakh, B., Reynolds, J., Melnikov, A., Kliushkina, N., Araya, C., Yan, S., and Reblitz-Richardson, O.: Captum: A unified and generic model interpretability library for PyTorch, https://doi.org/10.48550/arXiv.2009.07896, 16 September 2020.

Lundberg, S. M. and Lee, S.-I.: A Unified Approach to Interpreting Model Predictions, in: Advances in Neural Information Processing Systems, 2017.

Millan, R., Mouginot, J., Rabatel, A., and Morlighem, M.: Ice velocity and thickness of the world's glaciers, Nat. Geosci., 15, 124–129, https://doi.org/10.1038/s41561-021-00885-z, 2022.

**List of all relevant changes**

This list includes the changes that were already mentioned in the responses to the reviewers' comments.

L10: changed the sentence to "Glacier ice thickness is a fundamental variable required for modelling the evolution of a glacier."

L10-13: added "Ice thickness is the single most important input for modelling the dynamics of an ice mass because surface velocity is proportional to the fourth power of thickness (Cuffey and Paterson, 2010). Combined with surface elevation, it provides bed topography, also key for modelling flow."

L 13-14: Corrected citation: "In situ ice thickness measurements exist for only a fraction of the 215 000 glaciers in the world (Welty et al., 2020)"

L 15: change sentence to "There are physics-based and process-based approaches that aim to reconstruct glacier ice thicknesses from in situ data and ice dynamical considerations."

L24: added further reference to literature: "or ice thickness (Haq et al., 2021)."

L 35-39: elaborated on literature "PINNs and variations thereof were also already used for predicting ice flow (Jouvet and Cordonnier, 2023), inferring basal drag of ice streams (Riel et al., 2021) or ice shelf rheology (Wang et al., 2022; Iwasaki and Lai, 2023), for example. Cheng et al. (2024) built a unified framework involving a PINN to model ice sheet flow by enforcing momentum conservation

derived from the Shelfy-Stream Approximation. They apply their framework to a single glacier in Greenland to showcase the ability of the PINN to reconstruct ice thickness and basal friction simultaneously."

L 24: corrected sentence to "One advantage of machine learning approaches is their efficient optimization and evaluation compared to process-based models (Jouvet et al., 2022)."

L25: added "machine learning" to specify that we are talking about machine learning models that are purely data-driven

L34-35: changed wording from "boundary condition" to "internal condition": "Additional ground truth data can be used to compute a data loss that acts as an internal condition to constraining solutions to the PDE."

L 32: deleted "they [PINNs] are very data efficient" as data efficient could be misunderstood

L 44: added "As a proof of concept, we include all non-surging glaciers in Spitsbergen, Barentsøya, and Edgeøya in Svalbard to show that it is possible to use a PINN architecture for an entire region."

L71-94: rewrote Section 2.2:

"Glacier flow is the result of gravity-induced stresses on the ice. Friction between the ice and the glacier bed or sidewalls, friction between slower and faster-moving ice within the glacier, and gradients in longitudinal tension or compression encounter the gravitational stress (van der Veen, 2013).

The resulting ice movements depend on many factors, such as the physical properties of the ice like temperature, impurities, or density, and also conditions at the glacier bed (Jiskoot, 2011). From space, we can observe the surface velocity of glaciers. To infer thickness from mass conservation we would need to know the depth-averaged velocity.

There are models with different degrees of approximations to the full Navier-Stokes equations to describe ice flow. The simplest one, the shallow ice approximation (SIA) assumes lamellar flow, so the driving forces are entirely opposed by basal drag. It neglects lateral shear and longitudinal stresses and the rate factor A from Glen's flow law is taken to be constant with depth (van der Veen, 2013).

From this model, we can derive that the depth-averaged velocity relates to the surface velocity like $v = 0.8 v_s$ assuming the flow velocity at the base of the glacier is 0 (see Appendix A for derivation). However, basal velocity is unlikely to be 0.

The basal sliding velocity tightly relates to the properties of the glacier bed and complex interactions between water, sediment, and ice at the glacier bed (Cuffey and Paterson, 2010). Millan et al. (2022) introduced an empirical factor $\beta$ with $v_b = \beta v_s$ to account for contributions from basal sliding. They derive the factor from the ratio between surface slope and surface velocity.

If the ice velocity is entirely by slip along the glacier bed then $v_s = v_b = v$. Accordingly, we estimate the depth-averaged velocity to be within the bounds of

$$(l_{lower} + (1 - l_{lower}) \cdot \beta) \cdot v_s < v \leq v_s$$

where $l_{lower}$ acts as a parameterization for the vertical integration of the velocity and can be set between 0 and 1. Depending on the factor $\beta$ that lies between 0.1 and 1 the lower boundary is close to the defined $l_{lower}$ or closer to 1. For $\beta$ the lower boundary for the depth-averaged velocity equals the surface velocity."

L95-115: rewrote Section 2.3. Technical details can be found in the new Appendix B: PINN architecture and training.

"As already mentioned, a PINN consists of a neural network that is able to approximate the solution to a PDE (Karniadakis et al., 2021). A neural network, also sometimes called multi-layered perceptron, consists of layers of connected nodes, also called neurons, where the connections each have an associated weight. At each node, the weighted outputs from each node of the previous layer are passed through a non-linear activation function (Goodfellow et al., 2016). By minimizing a loss the weights of the network are updated to make accurate predictions.

In a PINN model the loss is given by the residual of the PDE we want to solve. In theory, PINNs only require input features that are needed to calculate the derivatives in the PDE (Raissi et al., 2018). In our work, we also provide the neural network with auxiliary data, that is related to glacier ice thickness but is not needed to solve the PDE. Therefore, we can exploit information from observable data as we would do it with a non-physics-aware neural network.

Additionally, we use a Fourier feature encoding layer as described by Tancik et al. (2020) preceding the neural network. A Fourier feature encoding layer maps input vector x to a higher dimensional feature space using $\gamma(x) = [\cos(2\pi Bx), \sin(2\pi Bx)]^T$. (4)

The embedding of spatial coordinates was originally developed to overcome spectral bias in neural networks and speed up convergence in the reconstruction of images. It enables the network to learn high-frequency functions in low-dimensional problem domains.

Figure 1 shows a schematic of the PINN model with its input features, outputs, and loss components. The exact architecture of the PINN is described in Appendix B The inputs to the model are vectors for each grid cell in the study region. They contain the spatial coordinates and surface velocities in x- and y-directions, and three $\beta$ values to correct for basal sliding in x- and y-direction and in the magnitude. Additionally, the vectors contain auxiliary data like elevation, slope, the grid cell's distance to the border of its glacier, and the area of the glacier it belongs to. Only the spatial coordinates get mapped to higher dimensional Fourier features."

L122: updated the loss function for the depth-averaged velocity loss to:

$$\mathcal{L}_{vel} = \begin{cases} (v_{\mathrm{s}} - \bar{v})^2 & \text{if } \bar{v} > v_{\mathrm{s}} \\ (v_{\mathrm{s}}(l_{\mathrm{lower}} + (1 - l_{\mathrm{lower}})\beta) - \bar{v})^2 & \text{if } \bar{v} < v_{\mathrm{s}}(l_{\mathrm{lower}} + (1 - l_{\mathrm{lower}})\beta) \\ 0 & \text{else} \end{cases} \quad \begin{array}{l} \text{with } \bar{v} \in \{\bar{v}_x, \bar{v}_y, \bar{v}_{\mathrm{mag}}\} \\ \text{and } \beta \in \{\beta_x, \beta_y, \beta_{\mathrm{mag}}\} \end{array}$$

L123: added: "As basal drag is most likely not the only drag the ice experiences, we decided to fix the lower bound as $l_{\mathrm{lower}} = 0.7$ in order to give more flexibility in the estimate."

L137: added explanation of labelled/unlabelled data: "We refer to the points with ice thickness measurements as labelled, whereas points without being referred to as unlabelled."

L168: added explanation of how OGGM provides the data we use: "OGGM reprojects and scales the data for each glacier to the glacier grids. We collect these data and transform the coordinates from the individual grids into a common projection."

L178: added "For each point, we compute three $\beta$ values from the surface velocities in the x- and y-direction and the magnitude of the surface velocity."

L201: added more information on auxiliary data: "Adding to the data that we need to impose the physics-aware losses, we also feed the network with extra information from auxiliary data as input features. We chose the features because they were easily available through OGGM and are related to the glacier's ice thicknesses. In Appendix E we analyze how each of the features impact the model output."

Table 1: added mean survey year for each of the test glaciers and the number of measurement points that were taken in total.

L215-217: added "Measurements on glaciers RGI60-07.00496 and RGI60-07.00497 are all from one survey, while the others are from multiple surveys carried out in different years."

Table 2: updated the results that changed after updating the calculation of the depth-averaged loss component

L225: to clarify we added: "The training and validation data are spatially correlated. Therefore, the in-sample evaluation of the model probably overestimates its performance."

L230: deleted comparison of variability to other ice thickness estimates as it does not add value: "This is low compared to the variation between the three physics-based models (Farinotti et al., 2019; Millan et al., 2022; van Pelt and Frank, 2024), with more than 0.70 variability for 90% of the points."

And added a conclusion of what the measurement of variability between the 7 LOGO CV models actually tells us: "As the in-sample validation scores of each model are also similar, we are confident that the method is robust to varying labelled data."

L231: exchanged "boundary conditions" for "target data" as we chose another wording in the beginning with "internal conditions"

Figure 4: adjusted the colour bar to better depict errors that exceed the min/max values

L232: Deleted "However, the model trained without thickness data of glacier RGI60-07.01482 overestimates its ice thickness." as this is not true for all points.

L242: added "Another example of that would be the comparison of performances on glaciers RGI60-07.00240 and RGI60-07.01481. They have similar measured ice thicknesses and RMSD scores but their MAPDs differ greatly." To underline that we need multiple metrics to measure performance

L280: added citation (Iwasaki and Lai, 2023)

L291: added example for measurement errors: "Ice thickness measurements from ice-penetrating radar, for example, are subject to errors due to varying density of glacier ice but also due to unknown thickness of snow and firn layers (Lindbäck et al., 2018)."

L297-298: added justification for statement: "which is also reported in other studies using multiple loss components in their PINNs (Iwasaki and Lai, 2023; Cheng et al., 2024)"

L299-200: added description of findings from our experiment: "In an experiment to test the importance of the loss components, we found that the relative importance is not very pronounced (see Appendix D). Therefore, we assume that the individual loss components are not optimally weighted in the reported model"

L207: changed the wording to clarify "Development of new optimization strategies"

L310-317: Rewrote the sentences to improve conciseness and readability

L318-321: added considerations about using in situ SMB data:

"Another option could be to use in situ mass balance data. This way, we circumvent the need for a mass balance model. The mass conservation loss would only be evaluated where data is available. This, however, would come with two restrictions. First, we would not be able to train the model in the entire study region. Secondly, also in situ mass balance data is not error-free. We would have to make a careful selection of the data to not introduce even more uncertainty."

L330-337: Mentioned other sources of uncertainty/simplifications in the model physics connecting to the estimate of depth-averaged velocity:

"We also want to mention that there are several processes affecting ice dynamics, especially in Svalbard, that are not very simplified or neglected in the model. One example is that our model assumes ice to be incompressible, when Svalbard glaciers actually have thick firn layers (Pälli et al., 2003). The varying density could introduce a non-negligible densification term in the mass balance Eq. (1).

Another example is the assumption of a temperature-independent creep coefficient A. Many glaciers in Svalbard are believed to be polythermal (Glasser, 2011). So the creep coefficient may vary within the ice, affecting the validity of our lower boundary for the estimate of depth-averaged velocity. However, the influence of these effects should carefully be weighed against the possibility of introducing errors if we decide to include better representations of these processes."

L338-340: added explanation of what we mean by "underconstrained problem":

"We only provide the model with the ice thickness measurements as a sort of internal condition, but we do not provide boundary conditions. Also, the depth-averaged velocity is only loosely constrained by a set of inequalities."

L345: changed wording for clarification "While this is technically easy to do, it comes at the cost of introducing uncertainties from approximating required parameters. We would need to assume ice viscosity and resistance from the bedrock, for example."

L346: added "In our view, the two elements that are most promising to improve the model performance, if revised, are the modelling of mass balance for Svalbard and the choice of surface velocity data for the estimation of depth-averaged velocities"

L353: added "Moreover, we have varying numbers of IPR measurements for the evaluation of each of the test glaciers as already mentioned in Sec. 4.1."

L365-367: added statement of what we judge as the most promising way to improve the model "Without changing the dataset we believe that optimizing the loss weights λ would have the biggest positive benefit, as the optimal configuration depends on the noise in the data Iwasaki and Lai (2023)."

L375: added "This serves as a proof of concept that physics-informed models can not only be applied to one single closed system but, together with auxiliary data, can make meaningful predictions for entire regions."

Appendices:

Changed the order of the first two Appendices so they fit with the main text

L389: added "A is, in general, dependent on the temperature of the ice, so $A = A(T)$."

L392: added "We further assume constant temperature within the ice so A does not depend on z"

L410-416: added Appendix B

"PINN architecture and training

[revised manuscript text omitted]

However, the results from the analysis are what we would expect from physical considerations. Therefore, it serves as a sanity check if the model is retrieving sensible correlations.

[Figure]

**Figure E1.** SHAP analysis: (a) Mean and standard deviation of the absolute SHAP values over all seven LOGO CV models. The values were first averaged for each model separately and then averaged for all seven models. (b) SHAP values for each datapoint by feature. The colour shows the relative value the feature takes for each datapoint. The SHAP values are calculated for the model trained without data from glacier RGI60-07.00240.

---

## Referee Report (RR1)

This work applies a Physics-Informed Neural Network (PINN) as a data-driven tool to estimate ice thickness across glaciers located on Spitsbergen in Svalbard. A physics-based loss function used in training the PINN is designed to penalize solutions diverging from a modified form of mass conservation. The neural network is also provided with a number of other data inputs, including surface velocity, surface slope, elevation, positional parameters, and values providing assumed relationships between surface and depth-averaged velocities. The authors explore their results using a cross-validation scheme designed to avoid problems with spatial correlation.

PINNs have seen an increasing number of uses within glaciology in the past few years. Their intrinsic ability to mix together known physics with poorly calibrated constants and sparse and/or noisy measurements make them an appealing modeling tool for underdetermined problems. This work is novel in training a single PINN over a heterogeneous domain consisting of multiple glaciers (effectively all of Spitsbergen) and in mixing physics-informed methods with purely data-driven methods.

The authors position their work as a proof of concept, largely demonstrating the ability to scale PINN-based methods to larger domains by mixing in non-physics-informed methods to produce continuous estimates in regions where no suitable boundary conditions exist.

In my view, the manuscript is much improved and offers a constructive contribution to the ongoing challenges of building useful models from sparse in-situ data. The availability of the authors' code and quality documentation adds to the value of the work.

**Framing of the work**
- In your response to the last round of reviewer comments, you made clear that you view your work as primarily a proof of concept. In that case, I'd suggest making clear in both the abstract and the introduction that your purpose with this article is to demonstrate a new technical method.
- Viewed as a technical proof of concept, I see what sets your work apart as primarily (a) the application of physics-informed techniques to a heterogenous spatial area spanning multiple catchments enabled by (b) a fusion of physics-informed and purely data-driven loss functions.

  If this is consistent with your view, I would suggest a few changes in your literature review:
  - Since your work fuses data-driven and physics-driven approaches, consider briefly reviewing non-physics-informed machine learning applications. Some that you might consider including:
    - Ice sheet scale thickness estimates: Leong and Horgan, 2020.
    - Glacier-scale SMB: Bolibar et al., 2019
    - Glacier-scale thickness: Haq et al., 2021

○ In addition to reviewing the Teisberg et al., 2021 application of PINNs to ice thickness mapping, also consider incorporating variational inference-based approaches such as Brinkerhoff et al., 2016.
○ I'm not sure I follow the comment about "without further consideration of bed properties" - perhaps expand on this if it's a key difference.
○ In framing your work as covering "an entire region," I think it's important to note that the significance of this is the heterogeneity of the region in question due to it being composed of multiple glaciers separate catchments. Notably, your domain is of roughly comparable size to the Rutford Ice Stream, Byrd Glacier, and the Amery Ice Shelf, the subjects of three of the works you cite. This is not to take away from what you are doing. It's just important to note that the significance is about the heterogeneity more than the pure spatial scale.

**Physical model**
● Section 2.2 is much clearer now.
● Line 79: I don't think that SIA necessarily implies neglecting the temperature dependence of the rate factor. See, for example, Larour et al., 2012. I suggest separating the constant A part out as a separate assumption you are adding.
● Equation 3 feels a bit abstract without knowing the selected values of l_lower and beta. Since these are specified later in the text, I suggest specifying their value/ranges briefly here (and referring readers to the appropriate sections below for details).

**PINN Evaluation**
● This is an extremely tricky issue, as there are not yet well established metrics for evaluating PINNs and this application is ever more challenging due to the combination of physics-informed and non-physics-informed loss functions. In general, the authors are doing a good job of discussing these nuances.
● I am not yet convinced by the analysis of the leave-one-out results and, in particular, in the overfitting conclusion. I would suggest evaluating the physics-informed loss function alone on the results of each of the 7 test glaciers when its ice thickness data was and wasn't included in the training (when it was or wasn't the left-out glacier, that is). Depending on the results of this…
  ○ Case 1: The physics-informed loss is about the same for each glacier with the thickness data left out or not.

  In this case, it seems that the model has found two different thickness maps which are roughly equally physically-plausible. This would be a very interesting result. I would argue this does not suggest "overfitting". The obvious follow-on question is: if you add in one point (or some very small amount of ice thickness data), does your model now predict the right thickness map?
  ○ Case 2: The physics-informed loss is much higher with the thickness data left out.

  In this case, I would agree with your "overfitting" assessment, but it's interesting

that your model is not finding a physically-plausible result without being guided by a few thickness measurements.

**Additional minor comments**

- Line 12: I find the opening claim that surface velocity is proportional to the fourth power of ice thickness to be potentially confusing. I assume you are referring to the paragraph following Eq. 8.36 in Cuffey and Paterson, which says that surface velocity is proportional to $H^4 * alpha^3$. Notably, however, ice thickness is inversely proportional to surface slope (Eq. 8.9). I would recommend softening the claim that ice thickness is the single most important input and citing a specific chapter of Cuffey and Paterson (8, I assume) to make it easier for readers to find the derivation you reference.
- Line 13: The importance of bed topography also needs a citation.
- Line 72: "encounter" doesn't make sense to me here. Perhaps you meant "counter" or "balance"?
- Line 78: "lamellar" -> "laminar"
- Line 111: Missing period after "Appendix B"
- Line 126 / Eq. 7: Equation 7 is not, in my opinion, a physics-aware constraint. I would argue that it is a heuristic smoothing regularization.
- Line 178: The units of the ratio here are a little confusing. I think the units here are (m/yr)/(m/m), which simplifies to m/yr. I recognize that is the same as $yr^{-1}*m$, however I would suggest either writing out the full un-simplified form or sticking to the more conventional $m*yr^{-1}$.

---

## Author Response (AR2)

Dear Ben,

Thank you again for giving us the opportunity to improve our manuscript. This letter combines our responses to the three reviewers' comments. As with last time, the comments are in black font, and our answers are in blue font.

We sincerely hope that the revised version meets your and the reviewers' expectations, and we look forward to your response.

Best wishes,

Viola Steidl and co-authors

**Point-by-point responses to comments from Reviewer 2**

Dear Reviewer #2,

Thank you for your comments and suggestions. We hope we can answer all your remarks and concerns to your satisfaction.

Thank you for helping us improve the manuscript.

Best wishes,

Viola Steidl and co-authors

This work applies a Physics-Informed Neural Network (PINN) as a data-driven tool to estimate ice thickness across glaciers located on Spitsbergen in Svalbard. A physics-based loss function used in training the PINN is designed to penalize solutions diverging from a modified form of mass conservation. The neural network is also provided with a number of other data inputs, including surface velocity, surface slope, elevation, positional parameters, and values providing assumed relationships between surface and depth-averaged velocities. The authors explore their results using a cross-validation scheme designed to avoid problems with spatial correlation.

PINNs have seen an increasing number of uses within glaciology in the past few years. Their intrinsic ability to mix together known physics with poorly calibrated constants and sparse and/or noisy measurements make them an appealing modeling tool for underdetermined problems. This work is novel in training a single PINN over a heterogeneous domain consisting of multiple glaciers (effectively all of Spitsbergen) and in mixing physics-informed methods with purely data-driven methods.

The authors position their work as a proof of concept, largely demonstrating the ability to scale PINN-based methods to larger domains by mixing in non-physics-informed methods to produce continuous estimates in regions where no suitable boundary conditions exist. In my view, the manuscript is much improved and offers a constructive contribution to the ongoing challenges of building useful models from sparse in-situ data. The availability of the authors' code and quality documentation adds to the value of the work.

Framing of the work

- In your response to the last round of reviewer comments, you made clear that you view your work as primarily a proof of concept. In that case, I'd suggest making clear in both the

abstract and the introduction that your purpose with this article is to demonstrate a new technical method.

Thank you for the comment. We added that this study serves as a proof of concept to the abstract, introduction and repeat it in the conclusion as well.

- Viewed as a technical proof of concept, I see what sets your work apart as primarily (a) the application of physics-informed techniques to a heterogenous spatial area spanning multiple catchments enabled by (b) a fusion of physics-informed and purely data-driven loss functions. If this is consistent with your view, I would suggest a few changes in your literature review:
  - Since your work fuses data-driven and physics-driven approaches, consider briefly reviewing non-physics-informed machine learning applications. Some that you might consider including:
    - Ice sheet scale thickness estimates: Leong and Horgan, 2020.
    - Glacier-scale SMB: Bolibar et al., 2019
    - Glacier-scale thickness: Haq et al., 2021

Thank you for the suggestions. We added Leong and Horgan to our literature review. Bolibar's work on SMB and Haq's work on ice thickness are both already in there.

- In addition to reviewing the Teisberg et al., 2021 application of PINNs to ice thickness mapping, also consider incorporating variational inference-based approaches such as Brinkerhoff et al., 2016.

Thank you for the suggestions. However, we are not sure if this would add nicely to our line of arguments, leading from the popular ice thickness maps from process-based models over to machine learning to model glaciers to physics-aware machine learning and its application to glaciers, and finally for ice thickness prediction. We would like to keep this introduction a bit slimmer and therefore, decided not to include this reference.

- I'm not sure I follow the comment about "without further consideration of bed properties" - perhaps expand on this if it's a key difference.

Agreed, this is not a key difference we want to focus on so we took this out.

- In framing your work as covering "an entire region," I think it's important to note that the significance of this is the heterogeneity of the region in question due to it being composed of multiple glaciers separate catchments. Notably, your domain is of roughly comparable size to the Rutford Ice Stream, Byrd Glacier, and the Amery Ice Shelf, the subjects of three of the works you cite. This is not to take away from what you are doing. It's just important to note that the significance is about the heterogeneity more than the pure spatial scale.

Thank you for your comment that this can be misunderstood. We changed the sentence to "… for a heterogenous region."

Physical model

- Section 2.2 is much clearer now.
- Line 79: I don't think that SIA necessarily implies neglecting the temperature dependence of the rate factor. See, for example, Larour et al., 2012. I suggest separating the constant A part out as a separate assumption you are adding.

Agreed, we divided this into two sentences.

- Equation 3 feels a bit abstract without knowing the selected values of l_lower and beta. Since these are specified later in the text, I suggest specifying their value/ranges briefly here (and referring readers to the appropriate sections below for details).

Thank you for your comment. As you said the values are specified in the following sentence so we would argue, there is no significant improvement in stating them the sentence before.

PINN Evaluation

- This is an extremely tricky issue, as there are not yet well established metrics for evaluating PINNs. This application is ever more challenging due to the combination of physics-informed and non-physics-informed loss functions. In general, the authors are doing a good job of discussing these nuances.
- I am not yet convinced by the analysis of the leave-one-out results and, in particular, in the overfitting conclusion. I would suggest evaluating the physics-informed loss function alone on the results of each of the 7 test glaciers when its ice thickness data was and wasn't included in the training (when it was or wasn't the left-out glacier, that is).

  Depending on the results of this...

  - Case 1: The physics-informed loss is about the same for each glacier with the thickness data left out or not. In this case, it seems that the model has found two different thickness maps which are roughly equally physically-plausible. This would be a very interesting result. I would argue this does not suggest "overfitting". The obvious follow-on question is: if you add in one point (or some very small amount of ice thickness data), does your model now predict the right thickness map?
  - Case 2: The physics-informed loss is much higher with the thickness data left out. In this case, I would agree with your "overfitting" assessment, but it's interesting that your model is not finding a physically-plausible result without being guided by a few thickness measurements.

Thank you very much for this comment. This is a good suggestion but it is not as easily implemented. We would have to filter for the specific glacier during the training which would add a substantial computational overhead to the algorithm. Leaving out the ice thickness measurements for the glacier is easier as this is done in data preparation before the training starts. To filter for a specific glacier during the training we would need to add the RGI ID to every training data point. But we definitely agree that this would be helpful to get deeper insights in the training mechanisms.

Additional minor comments

- Line 12: I find the opening claim that surface velocity is proportional to the fourth power of ice thickness to be potentially confusing. I assume you are referring to the paragraph following Eq. 8.36 in Cuffey and Paterson, which says that surface velocity is proportional to $H^4 * \alpha^3$. Notably, however, ice thickness is inversely proportional to surface slope (Eq. 8.9). I would recommend softening the claim that ice thickness is the single most important input and citing a specific chapter of Cuffey and Paterson (8, I assume) to make it easier for readers to find the derivation you reference.
  Thank you for the remark, we agree that this might be a bit too strong of a claim and therefore softened it

- Line 13: The importance of bed topography also needs a citation.
  Thanks for the remark, we added (van der Veen, 2013) as citation.
- Line 72: "encounter" doesn't make sense to me here. Perhaps you meant "counter" or "balance"?
  Agreed and adjusted.
- Line 78: "lamellar" -> "laminar"
  We went with the convention to use lamellar to describe glacier flow as in (van der Veen, 2013). Therefore, we would like to leave it as is.
- Line 111: Missing period after "Appendix B"
  Thanks, we corrected this.
- Line 126 / Eq. 7: Equation 7 is not, in my opinion, a physics-aware constraint. I would argue that it is a heuristic smoothing regularization.
  We agree that this can also be seen as a regularization as it is also done in non-physics-aware machine learning architectures. However, here it only applies to one of the model outputs instead of all of them and enhances the physical correctness.
- Line 178: The units of the ratio here are a little confusing. I think the units here are (m/yr)/(m/m), which simplifies to m/yr. I recognize that is the same as yr^-1*m, however I would suggest either writing out the full un-simplified form or sticking to the more conventional m*yr^-1.
  Thank you very much for the remark. Actually, it should be yr*m^-1, we corrected this.

van der Veen, C. J.: Fundamentals of Glacier Dynamics, Second edition., CRC Press, 2013.

**Point-by-point responses to comments from Reviewer 3**

Dear Reviewer #3,

We highly appreciate your thorough review of our manuscript and your suggestions to improve it. We hope we met your expectations with our revisions and answers to your comments.

Thank you very much and best wishes,

Viola Steidl and co-authors

**General comments**

The manuscript "Physics-aware Machine Learning for Glacier Ice Thickness Estimation: A Case Study for Svalbard" presents an interesting approach for reconstructing spatially-continuous glacier ice thickness from sparse in situ measurements with a neural network. The sparsity of the ice thickness measurements is mitigated by incorporating physical knowledge into the objective function, which acts as a regularizer on the neural network predictions. The authors demonstrate that the their method produces thickness estimates that are consistent with previous studies using physical models of ice flow.

After the first round of revisions, I believe that the authors have done a good job in addressing much of the concerns regarding the methodological details and factors that impact the accuracy of the thickness estimates. I particularly liked the inclusion of the SHAP analysis and the systematic analysis of the physics-aware loss functions.

However, based on the earlier reviews, it appears that a key remaining issue holding back this work is the lack of ground truth and the limited validation via comparison with previous ice thickness estimates. I would agree with the other reviewers that a synthetic experiment would be very beneficial in instilling confidence about the thickness estimates, especially since it appears that the estimates are systematically lower than the estimates from the previous approaches (Figure 5). While the authors state in their response that such an experiment would be a substantial effort, I believe that even simplified synthetics would go a long way towards clarifying the performance of the neural network.

Concretely, I would suggest that the authors generate 1D synthetics with a Blatter-Pattyn model with some random bed topography profiles. This can be done for a relatively small number of synthetic glaciers (~20) such that the number of velocity points is on the order of a few tens of thousands. A small fraction of ice thickness values can then be used as the auxiliary data. For the neural network features, since these are 1D, the features can be limited to the x-coordinate, slope, vx, and beta (and perhaps elevation if the modeled mass balance is elevation-dependent). These features (as well as the auxiliary mass balance) can be subjected to various amounts of noise. After the neural network is trained, then the thickness can be evaluated vs. the true thickness. Crucially, the thickness can also be evaluated against the reconstructed thickness using the approach of a model like Millan et al.'s. It may turn out that the previous approaches overestimate the thickness, which would be important to know. Overall, while I understand that such an experiment is not trivial by any means, it would really help increase the impact of this work and potentially highlight its technical advantages relative to previous approaches.

Thank you for this recommendation and also the suggestion on how to build a synthetic dataset. We decided to go for a simple approach, synthesizing data for a single glacier: We assumed basal stress in the direction of flow as the only stress component:

$$\tau_b = \rho * g * H * \alpha$$

and generated the surface velocity from

$$u_s = u_b + \frac{2A}{n+1} \tau_b^n H$$

.

The basal sliding velocity is set to zero for simplicity and the apparent mass balance *b* is calculated from

$$\frac{\partial H}{\partial t} + \nabla \cdot (\bar{v} H) = \dot{b}$$

.

Figure 1 shows the model results when training with and without physics-aware losses. The model performs significantly better if trained with physics-aware losses.

We did not evaluate Millan's model on our synthetic dataset as we felt this would exceed the scope of the manuscript. To us, this experiment serves as proof that the concept of using a physics-aware model delivers physically consistent ice thickness predictions, especially when compared to a machine-learning model that is not constrained by physics-aware losses. However, we would not like to draw any conclusions on whether another product is

overestimating or underestimating ice thickness.

[Figure]

*Figure 1* Predictions of models trained with and without physics-aware losses on 1D synthetic data

Other than the synthetics suggestion, I see no other major issues with this work. I've listed more specific comments below.

**Additional specific comments**

- The Fourier dimension and scale adds two more hyperparameters to the neural network design. How do these affect the performance of the reconstructions, and how are they chosen in this work?

We selected the hyperparameters for the Fourier embedding based on a series of non-exhaustive exploratory experiments rather than a full grid search to avoid computational costs. As described in (Tancik et al., 2020) choosing too many Fourier modes or the Fourier scale too big the predictions will be very grainy, while choosing them too small will not lead to the improvement of the mass conservation loss that we described in the answer letter to Reviewer #1.

- In Reviewer 1's comments, it was suggested to move the surface velocity data to the network outputs. Then, the data can be used as additional "auxiliary data" for constraining the predictions of the surface velocity. I think this is a good idea as it would provide a way to

smooth both the surface and depth-averaged velocities in a consistent manner. This approach could also be used to predict mass balance, constrained by more accurate in situ measurements. Thus, I think the authors should at least mention this as possible follow-up work.

If we understand correctly you suggest also predicting the surface velocity. We agree that this is a valid approach, which has also been followed by (Teisberg et al., 2021). We constructed the model in this way to explore a different approach, where we have the auxiliary data together with the physical constraints. Therefore, we wanted to provide the model with as many input fields as possible.

However, we agree that predicting also the surface velocity could lead to improved training of the model weights and is a configuration that could be explored in future work.

- I'm still confused why there should be three different beta values for the different velocity components. It seems to me that beta should be calculated from the velocity magnitudes, and then the same beta is used for all three components. This would keep the partitioning consistent.

We calculate three components for beta because we do a component-wise estimate of the depth-averaged velocity. Since the influence of basal sliding on the surface velocity might also be different depending on the direction we chose to simply calculate a beta value for the two components and the magnitude. To us this would be the most consistent way.

- Line 101: It's probably better to say "calculate the terms in the PDE" since there may be non-derivatives or products of derivatives depending on the PDE

Agreed and changed.

- Figure 1: In the caption, please add a little bit more description on the inputs, outputs, and losses here. What do the colors in the loss boxes correspond to?
 Agreed and improved.

- Line 130: Why not just predict log(H)

Thank you for this remark. This is actually another possibility of ensuring positive ice thickness that should be considered in future work.

- Line 153: I believe the use of the term "boundary condition" was changed to something like "internal condition" when referring to the ice thickness data, so this needs to be made consistent.

Thank you for the remark, we changed the wording to be consistent.

- Line 167: Does the varying grid resolution have an effect on generalization?

The grid resolution should not play a role in a PINN, but to be entirely sure, the LOGO CV could be run on a dataset where the grid resolution is kept the same for all the glaciers. We chose the varying grid resolutions to make sure that also smaller glaciers are represented well in the dataset with a big enough number of points.

- Line 175: Please state the size of the filter used for Gaussian smoothing (perhaps in units of fraction of average ice thickness)

The size of the smoothing filter depends on the grid resolution of the glaciers. From OGGM the smoothing window is set to 251/grid_resolution. We feel like this is a very technical detail of the data preparation that is set by OGGM and would not add to the clearness of the manuscript. Therefore, we would not like to include it in the main text of the manuscript. As all our code is publicly available, it is still well documented.

- Line 176: It's probably more accurate to say that $\beta$ is introduced to incorporate the effect of basal sliding on the measured surface velocity (instead of estimate)

Agreed and changed.

- Line 189: Also here, please state the size of the filter

Please see our answer above.

- Line 234: Since the in-sample and LOGO RMSD scores are significantly different, it would be useful to show a plot (probably in the Appendix) of the training performance (loss vs. training epoch) for the train and validation data. This way, we can get a better sense on the amount of over-fitting.

Figure 2 shows the mean loss curves for the data loss and the RMSD during the LOGO CV (validation RMSD is evaluated every 10 epochs only). The last point of the validation loss is the evaluation on the left-out test glacier. This plot again shows how important it is to have the LOGO CV as a random split for training and validation data would lead us to severely overestimate the performance of the model. All the ice thickness measurements are close to each other, so the model has no difficulty interpolating from one measurement point to the next. The overfitting can only be seen through the lens of a spatial split like we did it in the LOGO CV.

[Figure]

*Figure 2* Thickness data training and validation loss

- Line 289: It seems like it would be good to show a comparison of velocity fields between ~2010 and 2017-18 (if such data are available).

We agree that this would be interesting to see how much of an influence the different measurement dates of velocity and ice thickness can potentially have. However, extensive velocity data from earlier periods is hardly available due to high cloud coverage in this region. NASA's MEaSUREs project (https://its-live.jpl.nasa.gov/#access), for example, collects ice velocities for every year, but for the glaciers of Svalbard, there are only very few datapoints for years before 2015. However, since glaciers are melting rapidly (Hugonnet et al., 2021) we assume that the surface velocities show a significant change compared to 2010.

- Line 298: Even though the loss weights are held fixed, you should briefly discuss how these weights affect the final thickness estimates (or perhaps show an L-curve for the most important weight).
Thank you for the suggestion. We believe that with the study of the importance of physics-aware loss components in the Appendix, we have already explained how the weights of the loss components influence the predictive performance. Also, because the work is thought to be a proof-of-concept, we think that going more in-depth to explore the parameters of the model would exceed the scope of the manuscript.

- Figure 6: I would like to see one or two additional sentences in the caption that summarize the challenges or highlight the challenges that are most consequential for the proposed method.
We agree and added our key challenge to the caption.

- Line 355: This could be straightforwardly implemented by using thickness uncertainties as weights to the loss function

We agree that this would be one way to implement it. However, the uncertainty of the measurement is not added for every data point in GlaThiDa. Therefore, retrieving measurement uncertainty and annotating every ice thickness label would still be a great effort.

- Line 375-380: This paragraph is a bit too general and doesn't really add to the rest of the manuscript. Physics-aware machine learning has been around for quite some time now and has been applied to a wide variety of geophysical problems.
Thanks you for this comment, we agree that it might sound a bit general, so we altered it a bit. However, we want to encourage the readers to also take this work as an inspiration for other problems that might be solved with physics-aware machine learning. Therefore, we also provide the complete code of the model, to make the work not only reproducible but also reusable.

- Line 415: What about the batch size?

Thanks for the remark. The batch size is 8192 and we added it now.

- Appendix D title: Please state up front what metric you are measuring importance against (validation accuracy?)

Thanks again for the remark. Yes, it is measured on the validation set. We added the information to the manuscript.

**Point-by-point responses to comments from Reviewer 4**

Dear Reviewer #4,

Thank you for reviewing the manuscript, your comments and suggestions. We hope we can answer all your questions and concerns to your satisfaction.

Best wishes,

Viola Steidl and co-authors

The manuscript presents the development of a new physics-informed neural network (PINN) method to infer glacial ice thickness, based on a case study of Svalbard. While the description of the method in the initial manuscript was somewhat rough, the revised manuscript shows significant improvement. The other reviewers have addressed most of my questions, and the authors provided detailed responses. Below, I list some additional questions specific to the PINN method. I recommend the publication of this paper if the authors can address these points.

Questions:

1. The authors employ Fourier feature encoding within the network, which requires setting a hyperparameter B. From Tancik et al. (2020), we know that an incorrect choice of B can lead to overfitting in the network's predictions. Could the authors provide more details on how they determined the optimal value for B?

We agree that the choice of matrix B should be explained. We draw the entries for B from a gaussian distribution. The size of B and the standard deviation of the gaussian distribution are hyperparameters of the model. We selected the parameters from a non-exhaustive search, so they might not be optimal. The shape of B is [2,32] and the standard deviation is chosen as 10.0. We added this information to Appendix B.

2. On one hand, the authors apply Fourier feature encoding to capture high-frequency features in the output; on the other hand, they include a smoothness loss term in the loss function to regulate high derivatives. However, high-frequency functions typically exhibit high local derivatives, which appears somewhat contradictory to the use of a smoothness loss. Could the authors provide an explanation to clarify and justify these settings?

We agree that this seems to be counterintuitive. We do not have a rigorous explanation of this behaviour. Empirically, the Fourier feature encoding leads to an improved optimization of the mass conservation loss, as discussed in the response to Reviewer #1. The smoothness loss solely serves to fine-tune the training. There could well be a better configuration with fewer modes in the Fourier Feature encoding that does not need an additional smoothness loss.

3. In Appendix B (line 414), the authors state that they set the loss weights $\lambda_i \lambda_i$ to ensure all loss terms in the loss function are of the same order of magnitude. However, if the smoothness loss term is kept at the same magnitude as others, could this result in an over-smoothed thickness prediction?

Yes, if the smoothness loss is weighted too much, we assume we would have an oversmoothed prediction. However, in our case, the setting of the loss weights does not seem to lead to oversmoothed predictions.

4. Based on Figure 2, it appears that surface velocity and β-values are used solely as input features. However, both β and surface velocity also appear in the velocity loss term L_{vel} (Equation 6). It may be helpful to update Figure 2 to indicate that β-value and surface velocity are part of the data required for the loss function.

We agree that these connections should appear in the figure. As we would like to keep the figure clean, we decided to add this information to the caption.

5. As the authors mention, inferring glacial ice thickness is a highly under-constrained problem. Additionally, the sparsity of the measured thickness data complicates model validation. This may be beyond the scope of the current study, but it would be interesting if the authors could generate synthetic data to validate the robustness of the proposed PINN model.

We agree. We generated a synthetic 1D dataset to test the model performance on this dataset, as was suggested by Reviewer #3. Please refer to our answer letter to Reviewer #3 for a description of the outcome of this experiment.

6. Did the authors conduct experiments with different weight initializations or network structures to assess whether the PINN training converges to a unique thickness inference? If so, could they provide the standard deviation of these tests?

We did not conduct experiments with different weight initializations but we conducted a non-exhaustive search for the best model parameters considering different numbers of layers within the network, for example. We did not measure the standard deviation of the thickness predictions.

7. Given that the authors used various input features in the training, I suggest adding more detail on the input layers in Appendix B to help readers better understand the network structure used.

The input layer for the model is a linear layer with 256 neurons just like the other layers in the model. Therefore, there is no specific description of the input layer.

**List of relevant changes**

L. 7: added "proof-of-concept" to describe the experiments of the manuscript

L. 10: deleted "single" to soften the claim that ice thickness is an important input for modelling glacier dynamics

L. 13: added citation

L. 23: added citation

L.41: deleted "without further consideration of bed properties" as this detail was not adding to the clarity of the manuscript

L. 46: exchanged "entire" for "heterogenous" to better describe the achievement of this study

L. 73: exchanged "encounter" with "balance"

Figure 1: extended the caption with the sentences: "The physics-aware losses are in purple boxes. The Data loss in the blue box is the only loss depending on ice thickness measurement data. Surface velocity and $\beta$-values also add to the physics-aware losses. The connection is not shown to increase readability."

L. 133: exchanged "boundary condition" for "internal condition"

L. 141: added: "We tested a slim version of the PINN model on a one-dimensional data set of a single glacier. The results are given in App. C. The experiment shows the added value of introducing physics-aware loss components."

L. 155: exchanged "boundary condition" for "internal condition"

L. 179exchanged "estimate" for "incorporate"

Figure 6: extended the caption with: "Challenges for PINNs in a real-world setting like the prediction of glacier ice thickness. The separate realms are interfering with each other, complicating the optimization of the model. Weighting of the physical constraints could have the biggest positive benefit."

L. 380: exchanged "physical law or condition provides a strong constraint" with " physical law and multiple conditions provide constraints"

Appendix B: added description of the parameters for the Fourier feature encoding layer and the value chosen as batch size

Appendix C: added experiment on synthetic data.

Appendix E: added "calculating a relative RMSD on the validation set"